# Collaborative Video Diffusion: Consistent Multi-video Generation with Camera Control

**Zhengfei Kuang*[1]**  **Shengqu Cai*[1]**  **Hao He[2]**  **Yinghao Xu[1]**
**Hongsheng Li[2]**  **Leonidas Guibas[1]**  **Gordon Wetzstein[1]**

[1]Stanford Univerity  [2]CUHK

## Abstract

Research on video generation has recently made tremendous progress, enabling high-quality videos to be generated from text prompts or images. Adding control to the video generation process is an important goal moving forward and recent approaches that condition video generation models on camera trajectories make strides towards it. Yet, it remains challenging to generate a video of the same scene from multiple different camera trajectories. Solutions to this multi-video generation problem could enable large-scale 3D scene generation with editable camera trajectories, among other applications. We introduce collaborative video diffusion (CVD) as an important step towards this vision. The CVD framework includes a novel *cross-video synchronization module* that promotes consistency between corresponding frames of the same video rendered from different camera poses using an epipolar attention mechanism. Trained on top of a state-of-the-art camera-control module for video generation, CVD generates multiple videos rendered from different camera trajectories with significantly better consistency than baselines, as shown in extensive experiments. Project page: https://collaborativevideodiffusion.github.io/.

## 1 Introduction

With the impressive progress of diffusion models [22, 51, 28, 44, 46, 43], video generation has significantly advanced [12, 17, 26, 5, 6, 15, 64, 23, 32], with a transformative impact on digital content creation workflows. Recent models like SORA [7] exhibit the ability to generate long high-quality videos with complex dynamics. Yet, these methods typically leverage text or image inputs to control the generation process and lack precise control over content and motion, which is essential for practical applications. Prior efforts explore the use of other input modalities, such as flow, keypoints, and depths, and develop novel control modules to incorporate these conditions effectively, enabling precise guidance of the generated contents [60, 63, 67, 26, 56, 9]. Despite these advancements, these methods still fail to provide camera control to the video generation process.

Recent works have started to focus on camera control using various techniques, such as motion LoRAs [25, 17] or scene flows [63, 67]. Some representative works such as MotionCtrl [60] and CameraCtrl [18] offer more flexible camera control by conditioning the video generative models on a sequence of camera poses, showing the feasibility of freely controlling the camera movements of videos. However, these methods are limited to single-camera trajectories, leading to significant inconsistencies in content and dynamics when generating multiple videos of the same scene from different camera trajectories. Consistent multi-video generation with camera control is desirable in many downstream applications, such as large-scale 3D scene generation. Training video generation models for consistent videos with different camera trajectories, however, is very challenging, partly due to the lack of large-scale multi-view dynamic in-the-wild scene data.

38th Conference on Neural Information Processing Systems (NeurIPS 2024).

In this paper, we introduce CVD, a plug-and-play module capable of generating videos with different camera trajectories sharing the same underlying content and motion of a scene. CVD is designed on a collaborative diffusion process that generates consistent pairs of videos with individually controllable camera trajectories. Consistency between corresponding frames of a video is enabled using epipolar attention, introduced by a learnable *cross-view synchronization module*. To effectively train this module, we propose a new pseudo-epipolar line sampling scheme to enrich the epipolar geometry attention. Due to the shortage of large-scale training data for 3D dynamic scenes, we propose a *hybrid training* scheme where multi-view static data from RealEstate10k [68] and monocular dynamic data from WebVid10M [1] are utilized to learn camera control and motion, respectively. To our knowledge, CVD is the first approach to generate multiple videos with consistent content and dynamics while providing camera control. Through extensive experiments, we demonstrate that CVD ensures strong geometric and semantic consistencies, significantly outperforming relevant baselines. We summarize our contributions as follows:

- To our knowledge, our CVD is the first video diffusion model that generates multi-view consistent videos with camera control;
- We introduce a novel module called the *Cross-Video Synchronization Module*, designed to align features across diverse input videos for enhanced consistency;
- We propose a new collaborative inference algorithm to extend our video model trained on video pairs to arbitrary numbers of video generation;
- Our model demonstrates superior performance in generating multi-view videos with consistent content and motion, surpassing all baseline methods by a significant margin.

## 2   Related Work

**Video Diffusion Models.**   Recent efforts in training large-scale video diffusion models have enabled high-quality video generation [15, 6, 23, 21, 17, 7, 12, 50]. Video Diffusion Model [23], utilizes a 3D UNet to learn from images and videos jointly. With the promising image quality obtained by text-to-image (T2I) generation models, such as StableDiffusion [44], many recent efforts focus on extending pretrained T2I models by learning a temporal module. Align-your-latents [6] proposes to inflate the T2I model with 3D convolutions and factorized space-temporal blocks to learn video dynamics. Similarly, AnimateDiff [17] builds upon StableDiffusion [44], adding a temporal module after each fixed spatial layer to achieve plug-and-play capabilities that allow users to perform personalized animation without any finetuning. Pyoco [15] proposes a temporally coherent noise strategy to effectively model temporal dynamics. More recently, SORA [7] shows a great step towards photo-realistic long video generation by utilizing space-time diffusion with a transformer architecture.

**Controllable Video Generation.**   The ambiguity of textual conditions often results in weak control for text-to-video models (T2V). To provide precise guidance, some approaches utilize additional conditioning signals such as depth, skeleton, and flow to control the generated videos [12, 59, 26, 48, 27, 8, 53]. Recent efforts like SparseCtrl [65] and SVD incorporate images as control signals for video generation. To further control motions and camera views in the output video, DragNUWA [63] and MotionCtrl [60] inject motion and camera trajectories into the conditioning branch, where the former uses a relaxed version of optical flow as stroke-like interactive instruction, and the later directly concatenate camera parameters as additional features. CameraCtrl [18] proposes to over-parameterize the camera parameters using Plücker Embeddings [39] and achieves more accurate camera conditioning. Alternatively, AnimateDiff [17] trains camera-trajectory LoRAs [25] to achieve viewpoint movement conditioning, while MotionDirector [67] also utilizes LoRAs [25] but to overfit to specific appearances and motions to gain their decoupling.

**Multi-View Image Generation.**   Due to the lack of high-quality scene-level 3D datasets, a line of research focuses on generating coherent multi-view images. Zero123 [35] learns to generate novel-view images from pose conditions, and subsequent works extend it to multi-view diffusion [11, 36, 37, 49, 54, 55, 62, 31] for better view consistency. However, these methods are only restricted to objects and consistently fail to generate high-quality large-scale 3D scenes. MultiDiffusion [3] and DiffCollage [66] facilitates 360-degree scene image generation, while SceneScape [14] generates zooming-out views by warping and inpainting using diffusion models. Similarly, Text2Room [24] generates multi-view images of a room, where the images can be projected via

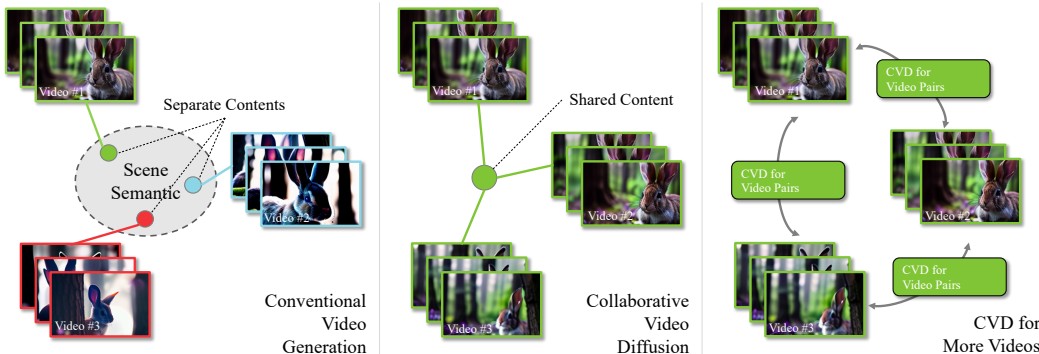

Figure 1: **An illustration of pairwise collaborative video generation**. Existing video diffusion models generate videos separately, which may result in inconsistent frame contents (e.g., geometries, objects, motions) across videos (*Left*); Collaborative video generation aims to produce videos sharing the same underlying content (*Middle*); In this work, we train our model on video pair datasets, and extend it to generate more collaborative videos (*Right*).

depths to get a coherent room mesh. DiffDreamer [9] follows the setups in Infinite-Nature [34, 33] and iteratively performs projection and refinement using a conditional diffusion model. A recent work, PoseGuided-Diffusion [56], performs novel view synthesis from a single image by training and adding an epipolar line bias to its attention masks on multi-view datasets with camera poses provided (RealEstate10k [68]). However, this method by construction does not generalize to in-the-wild or dynamic scenes, as its prior is solely learned from well-defined static indoor data.

A comprehensive survey of recent advances in diffusion models for visual computing is provided by Po et al. [40].

## 3 Collaborative Video Generation

Conventionally, video diffusion models (VDMs) aim to generate videos from randomly sampled Gaussian noise with multiple denoising steps, given conditions such as text prompts, frames, or camera poses. Specifically, let $v_0 \sim q_{\text{data}}(v)$ be a data point sampled from the data distribution; the forward diffusion process continuously adds noises to $v_0$ to get a series of $v_t, t \in 1, ..., T$ until it becomes Gaussian noise. Using the reparameterization trick from Ho et al.[22], the distribution of $v_t$ can be represented as $q(v_t \mid v_0) = \mathcal{N}(v_t; \sqrt{\bar{\alpha}_t}v_0, (1 - \bar{\alpha}_t)I)$, where $\bar{\alpha}_t \in (0, 1]$ are the noise scheduling parameters, which are monotonously increasing, and $\bar{\alpha}_T = 1$. The video diffusion model, typically denoted as $p_\theta(v_{t-1}|v_t)$, is a model parameterized by $\theta$ that is trained to estimate the backward distribution $q(v_{t-1}|v_t, v_0)$. According to Ho et al. [22], the optimization of $p_\theta(v_{t-1}|v_t)$ results in minimizing the following loss function:

$$\mathcal{L} = \mathbb{E}_{\epsilon, v_0, t, c}\|\epsilon - \epsilon_\theta(v_t, t, c)\|^2, \tag{1}$$

where $v_t = \sqrt{\bar{\alpha}_t}v_0 + (1 - \bar{\alpha}_t)\epsilon$ is the noisy video feature generated from $v_0$ and a random sampled Gaussian noise $\epsilon$, $\epsilon_\theta(v_t, t)$ is the noise prediction of the VDM, and $c$ is the video condition. During inference time, one can start from a normalized Gaussian noise $v_T \sim \mathcal{N}(0, I)$ and apply the noise prediction model $\epsilon_\theta(v_t, t)$ multiple times to denoise it until $v_0$.

Empowered by readily available large-scale video datasets, many state-of-the-art VDMs have successfully shown ability to produce temporally consistent and realistic videos [23, 5, 7, 17, 26, 21, 12, 6, 15]. However, one of the key drawbacks of all these existing methods is the inability to generate consistently coherent multi-view videos. As Fig. 1 shows, videos generated from a VDM under the same textual conditions exhibit content and spatial arrangement disparities. One can use inference-stage tricks, such as extended attention [8], to increase the semantic similarities between the videos, yet this does not address the problem of structure consistency. To address this issue, we introduce a novel objective for VDMs to generate multiple structurally consistent videos simultaneously given certain semantic conditions and dub it *Collaborative Video Diffusion* (CVD).

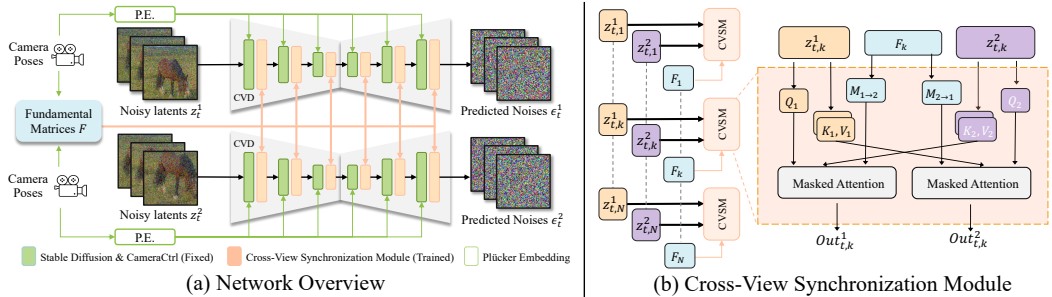

(a) Network Overview

(b) Cross-View Synchronization Module

Figure 2: **Architecture of collaborative video diffusion**. *Left*: The model takes two (or more) noisy video features and camera trajectories as input and generates the noise prediction for both videos. Note that the image autoencoder of Stable Diffusion is omitted here; *Right*: Our *Cross-View Synchronization Module* takes the same frames from the two videos along with the corresponding fundamental matrix as input, and applies a masked cross-view attention between the frames.

In contrast to conventional video diffusion models, CVD seeks to find an arbitrary number of videos $v^i, i \in 1, ..., M$ that comply with the unknown data distribution $q_{\text{data}}(v^{1,...,M})$ given separate conditions $c^{1,...,M}$. Similarly, the CVD model can be represented as $p_\theta(v^{1,...,M}|c^{1,...,M})$. An example includes multi-view videos synchronously captured from the same dynamic 3D scene. Similarly, the loss function for a collaborative video diffusion model is defined as:

$$\mathcal{L}_{\text{CVD}} = \mathbb{E}_{\epsilon^{1,...,M}, v_0^{1,...,M}, t, c} \| \epsilon^{1,...,M} - \epsilon_\theta(v_t^{1,...,M}, t, c^{1,...,M}) \|^2. \tag{2}$$

In practice, however, the scarcity of large-scale multi-view video data prevents us from directly training a model for an arbitrary quantity of videos. Therefore, we build our training dataset of consistent video pairs (i.e., $M = 2$) from existing monocular video datasets, and train the diffusion model to generate pairs of videos sharing the same underlying contents and motions (see details in Secs. 4.1 and 4.2). Our model is designed to accommodate any number of input video features, however, and we develop an inference algorithm to generate an arbitrary number of videos from our pre-trained pairwise CVD model (see Sec. 4.3).

## 4 Collaborative Video Diffusion with Camera Control

We seek to build a diffusion model that takes a text prompt $y$ and a set of camera trajectories $cam^{1,...,M}$ and generates the same number of collaborative videos $v^{1,...,M}$. To ease the generation of consistent videos, in this work we train our model with video pairs ($M = 2$), we make the assumption that the videos are synchronized (i.e., corresponding frames are captured simultaneously), and set the first pose of every trajectory to be identical, forcing the first frame of all videos to be the same.

Inspired by [18, 17], our model is designed as an extension of the camera-controlled video model CameraCtrl [18]. As shown in Fig. 2, our model takes two (or more) noisy video feature inputs and generates the noise prediction in a single pass. The video features pass through the pretrained weights of CameraCtrl and are synchronized in our proposed *Cross-View Synchronization Modules* (Sec. 4.1). The model is trained with two different datasets: RealEstate10K [68], which consists of camera-calibrated video on mostly static scenes, and WebVid10M [1], which contains generic videos without poses. This leads to our two-phase training strategy introduced in Sec. 4.2. The learned model can infer arbitrary numbers of videos using our proposed inference algorithm, which will be described in Sec. 4.3.

### 4.1 Cross-View Synchronization Module

State-of-the-art VDMs commonly incorporate various types of attention mechanisms defined on the spatial and temporal dimension: works such as AnimateDiff [17], SVD [5], LVDM [19] disentangles space and time and applies separate attention layers; the very recent breakthrough SORA [7] processes both dimensions jointly on its 3D spatial-temporal attention modules. Whilst the operations defined on the spatial and temporal dimensions bring a strong correlation between different pixels of different frames, capturing the context between different videos requires a new operation: cross-video attention.

Thankfully, prior works [10, 8] have shown that the extended attention technique, i.e., concatenating the key and values from different views together, is evidently efficient for preserving identical semantic information across videos. However, it refrains from preserving the structure consistency among them, leading to totally different scenes in terms of geometry. Thus, inspired by [56], we introduce the *Cross-View Synchronization Module* based on the epipolar geometry to shed light on the structure relationship between cross-video frames during the generation process, aligning the videos towards the same geometry.

Fig. 2 demonstrates the design of our cross-view module for two videos. Taking a pair of feature sequences $\mathbf{z}_{1,...,N}^1, \mathbf{z}_{1,...,N}^2$ of $N$ frames as input, our module applies a cross-video attention between the same frames from the two videos. Specifically, we define our module as:

$$out_k^1 = \texttt{ff}(\texttt{Attn}(\mathbf{W}_Q\mathbf{z}_k^1, \mathbf{W}_K\mathbf{z}_k^2, \mathbf{W}_V\mathbf{z}_k^2, \mathcal{M}_k^{1,2})), \quad \forall k \in 1, ..., N, \tag{3}$$

$$\mathcal{M}_k^{1,2}(\mathbf{x}_1, \mathbf{x}_2) = \mathbf{1}(\mathbf{x}_2^T\mathbf{F}_k^{1\rightarrow 2}\mathbf{x}_1 < \tau_{\text{epi}}) \tag{4}$$

where $k$ is the frame index, $\mathbf{W}_Q, \mathbf{W}_K, \mathbf{W}_V$ are the query, key, value mapping matrices, $\mathcal{M}$ is the attention mask, $\texttt{Attn}(Q, K, V, \mathcal{M})$ is the attention operator introduced from the Transformer [58], $\texttt{ff}$ is the feed-forward function and $\mathbf{F}_k^{1\rightarrow 2}$ is the fundamental matrix between $cam_k^1$ and $cam_k^2$. The attention mask $\mathcal{M}$ between any two pixels $\mathbf{x}_1, \mathbf{x}_2$ is determined by the epipolar distance between $\mathbf{x}_1$ and $\mathbf{x}_2$, i.e. the shortest distance between $\mathbf{x}_1$ and the epipolar line of $\mathbf{x}_2$ in $\mathbf{x}_1$'s frame, which is set to 1 if the epipolar distance is smaller than a given threshold $\tau_{\text{epi}}$ (set to 3 in all of our experiments) and vise versa. The outputs of these modules are used as residual connections with corresponding original inputs to ensure no loss of originally learned signals. The key insight of this module is as the two videos are assumed to be synchronized to each other, the same frame from the two videos is supposed to share the same underlying geometry and hence can be correlated by their epipolar geometry defined by the given camera poses. For the first frames where the camera poses are set to be identical since the fundamental matrix is undefined here, we generate pseudo epipolar lines for each pixel with random slopes that go through the pixels themselves. In the scenario where multi-view datasets are available, the modules can be further adapted to more videos by extending the cross-view attention from 1-to-1 to 1-to-many. Our study shows that epipolar-based attention remarkably increases the geometry integrity of the generated video pairs.

## 4.2 Hybrid Training Strategy from Two Datasets

Considering the fact that there is no available large-scale real-world dataset for video pairs, we opt to make use of the two popular monocular datasets, RealEstate10K [68] and WebVid10M [1], to develop a hybrid training strategy for video pair generation models.

**RealEstate10K with Video Folding.** The first phase of the training involves RealEstate10K [68], a dataset consisting of video clips capturing mostly static indoor scenes and corresponding camera poses. We sample video pairs by simply sampling subsequences of $2N - 1$ frames from a video in the dataset, then cutting them from the middle and reversing their first parts to form synchronized video pairs. In other words, the subsequences are folded into two video clips sharing the same starting frame.

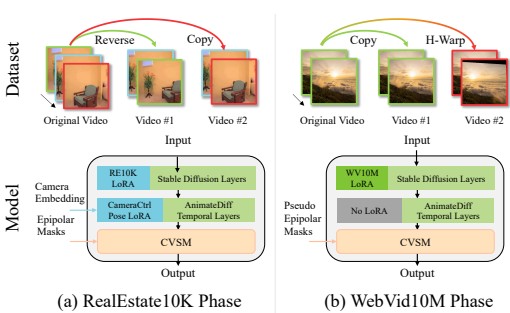

Figure 3: **Two-Phase Hybrid Training.** We use different data processing schemes to handle the two datasets (*Top*) and apply separate model structures to train in corresponding phases (*Bottom*).

**WebVid10M with Homography Augmentation.** While RealEstate10K [68] provides a decent geometry prior, training our model only on this dataset is not ideal since it does not provide any knowledge regarding dynamics and only contains indoor scenes. On the other hand, WebVid10M, a large-scale video dataset, consists of all kinds of videos and can be used as a good supplement to RealEstate10K. To extract video pairs, we clone the videos in the dataset and then apply random homography transformations to the clones. Nonetheless, The WebVid10M dataset contains no camera information, making it unsuitable for camera-conditioned model training. To address this problem,

we propose a two-phase training strategy to adapt both datasets (with or without camera poses) for the same model.

**Two-Phase Training.** As previously mentioned, our model is built upon the existing camera-controlled VDM CameraCtrl [18]. It is an extended version of AnimateDiff [17] that adds a pose encoder and several pose feature injectors for the temporal attention layers to the original model. Both AnimateDiff [17] and CameraCtrl [18] are based on Stable Diffusion [44]. This implies that they incorporate the same latent space domain, and thus, it is feasible to train a module that can be universally adapted. Therefore, our training procedure follows a two-phase scheme Fig. 3 shows. Specifically, we build a hybrid dataset that combines the data from both sources. Then in each training iteration, if the training data is from RealEstate10K, we use CameraCtrl with LoRA fine-tuned on RealEstate10K as the backbone and applying the ground truth epipolar geometry in the cross-video module. Otherwise, we use AnimateDiff with LoRA fine-tuned on WebVid10M as the backbone, and apply the pseudo epipolar geometry (the same strategy used for the first frames in RealEstate10K dataset) in the cross-video module. The two training phases are applied alternatively to the same instance of CVSM in a single training procedure. Experiments show that the hybrid training strategy greatly helps the model generate videos with synchronized motions and great geometry consistency.

### 4.3 Towards More Videos

With the CVD trained on video pairs, during inference, we can generate multiple videos (for example, $M$ videos where $M > 2$) that share consistent content and motions. To achieve that, we start from $M$ individual gaussian noise maps and denoise them in multiple steps. At each denoising step $t$, we select $P$ feature pairs $\mathcal{P} = \{\boldsymbol{v}_t^{i_1,j_1}, \boldsymbol{v}_t^{i_2,j_2}, ..., \boldsymbol{v}_t^{i_P,j_P} \mid i_{1,...,P}, j_{1,...,P} \in 1, ..., M\}$ among all $M$ video features. We then use the trained network to predict the noise of each feature pair, and averaging them w.r.t. each video feature. That is, the output noise for the $i$th video feature is defined as: $\boldsymbol{\epsilon}_{out}(\boldsymbol{v}_t^i) = \mathrm{Avg}_{\boldsymbol{v}^{i,j} \in \mathcal{P}}(\boldsymbol{\epsilon}_\theta^i(\boldsymbol{v}_t^{i,j}, t, cam^{i,j}))$, where $\boldsymbol{\epsilon}_\theta^i(\boldsymbol{v}_t^{i,j}, t, cam^{i,j})$ is the noise prediction for $\boldsymbol{v}_t^i$ given the video pair input $\boldsymbol{v}_t^{i,j}$. For pair selection, we propose the following strategies:

- *Exhaustive Strategy*: Select all $M(M-1)/2$ pairs.
- *Partitioning Strategy*: Randomly divide $M$ noisy video inputs into $\frac{M}{2}$ pairs.
- *Multi-Partitioning Strategy*: Repeat the Partitioning Strategy multiple times and combine all selected pairs.

The exhaustive strategy has a higher computational complexity of $O(M^2)$ compared to the partitioning one ($O(M)$) but covers every pair among $M$ videos and thus can produce more consistent videos. The multi-partitioning strategy, on the other hand, is a trade-off between the two strategies. We also embrace the recurrent denoising method introduced by Bansal et al. [2] that does multiple recurrent iterations on each denoising timestep. We provide the pseudo-code of our inference algorithm and detailed mathematical analysis in our supplementary.

## 5 Experiments

### 5.1 Qualitative Results

#### 5.1.1 Comparison with Baselines

Qualitative comparisons are shown in Fig. 4. Following our quantitative comparisons in Sec. 5.2, we compare against CameraCtrl [18] and its combination with SparseCtrl [16], MotionCtrl [60] and its combination with SVD [5]. The results indicate our method's superiority in aligning the content within the videos, including dynamic content such as lightning, waves, etc. More qualitative results are provided in our supplemental material and video.

#### 5.1.2 Additional results for arbitrary views generation

We also show the results of arbitrary view generation shown in Fig. 5. Using the algorithm introduced in Sec. 4.3, our model can generate groups of different camera-conditioned videos that share the same contents, structure, and motion. Please refer to our supplementary video for animated results.

## 5.2 Quantitative Results

We compare our model with two state-of-the-art camera-controlled video diffusion models for quantitative evaluation: CameraCtrl [18] and MotionCtrl [60]. Both of the two baselines are trained on the RealEstate10K [68] for camera-controlled video generation. We conduct the following experiments to test the geometric consistency, semantic consistency, and video fidelity of all models:

Table 1: **Quantitative Results on Geometry Consistency.** Following SuperGlue [47], we report the area under the cumulative error curve (AUC) of the predicted camera rotation and translation under certain thresholds ($5°, 10°, 20°$), and the precision (P) and matching score (MS) of the Super-Glue correspondences. We feed the models with prompts from RealEstate10K [68] (RE10K) and WebVid10M [1] (WV10M) in two experiments separately. For RealEstate10K scenes, we also run SuperGlue on the original RealEstate10K [68] frames as reference. Our model achieves the highest scores on all metrics compared to baselines.

| Scenes | Methods | Rot. AUC ↑ (@5°/10°/20°) | Trans. AUC ↑ (@5°/10°/20°) | Prec. ↑ | M-S. ↑ |
|---|---|---|---|---|---|
| RE10K | Reference | 61.4 / 77.2 / 87.8 | 6.9 / 17.5 / 41.0 | 60.2 | 36.5 |
| | CameraCtrl [18] | 34.8 / 55.2 / 72.4 | 2.3 / 6.6 / 17.0 | 50.8 | 27.3 |
| | MotionCtrl [60] | 49.0 / 68.0 / 81.2 | 3.4 / 10.2 / 25.0 | 64.6 | 38.9 |
| | Ours | **55.5 / 71.8 / 83.3** | **5.6 / 15.9 / 33.2** | **76.9** | **42.3** |
| WV10M | CameraCtrl [18]+SparseCtrl [16] | 6.2 / 14.3 / 25.8 | 0.5 / 1.7 / 4.7 | 16.5 | 5.4 |
| | MotionCtrl [60]+SVD [5] | 12.2 / 28.2 / 48.0 | 1.2 / 4.9 / 13.5 | 23.5 | 12.8 |
| | Ours | **25.2 / 40.7 / 57.5** | **3.7 / 9.6 / 19.9** | **51.0** | **23.5** |

Table 2: **Quantitative Results for semantic & fidelity metrics.** The semantic metrics are evaluated on WebVid10M [1] and the fidelity metrics are performed on RealEstate10k [68]. As shown in the table, our method is better than or on par with all prior work regarding semantic matching with the prompt, cross-video consistency, and frame fidelity.

| | Semantic Consistency | | Fidelity | | |
| | CLIP-T ↑ | CLIP-F ↑ | FID ↓ | KID ↓ | FVD ↓ |
|---|---|---|---|---|---|
| MotionCtrl [60]+SVD [5] | - | 0.81 | - | - | - |
| CameraCtrl [18] | 0.28 | 0.79 | **32.10** | 0.79 | **277** |
| AnimateDiff [17]+SparseCtrl [16] | 0.29 | 0.86 | 51.97 | 1.86 | 327 |
| CameraCtrl [18]+SparseCtrl [16] | 0.29 | 0.85 | 61.68 | 2.47 | 430 |
| Ours | **0.30** | **0.93** | 32.90 | **0.61** | 285 |

**Per-video geometric consistency on estate scenes.** Following CameraCtrl [18], we first test the geometry consistency across the frames in the video generated from our model, using the camera trajectories and text prompts from RealEstate10K [68] (which mostly consists of static scenes). Specifically, we first generate 1000 videos from randomly sampled camera trajectory pairs (two camera trajectories with the same starting transformation) and text captions. All baselines generate one video at a time; our model generates two videos simultaneously. For each generated video, we apply the state-of-the-art image matching algorithm SuperGlue [47] to extract the correspondences between its first frame and following frames and estimate their relative camera poses using the RANSAC [13, 42] algorithm. To evaluate the quality of correspondences and estimated camera poses, we adopt the same protocol from SuperGlue [47], which 1) evaluates the poses by the angle error of their rotation and translation and 2) evaluates the matched correspondences by their epipolar error (i.e., the distance to the ground truth epipolar line). The results are shown in Tab. 1, where our model significantly outperforms all baselines. More details are provided in our supplementary materials.

**Cross-video geometric consistency on generic scenes.** Aside from evaluating the consistency between frames in the same video, we also test our model's ability to preserve the geometry information across different videos. To do that, we randomly sample 500 video pairs (1000 videos in total) using camera trajectory pairs from RealEstate10K [68] and text prompts from WebVid10M's captions [1]. To the best of our knowledge, there is no available large video diffusion model that is designed to generate multi-view consistent videos for generic scenes. Hence, we modify the CameraCtrl [18] and MotionCtrl [60] to generate video pairs as baselines. Here, we use the text-to-video version of each

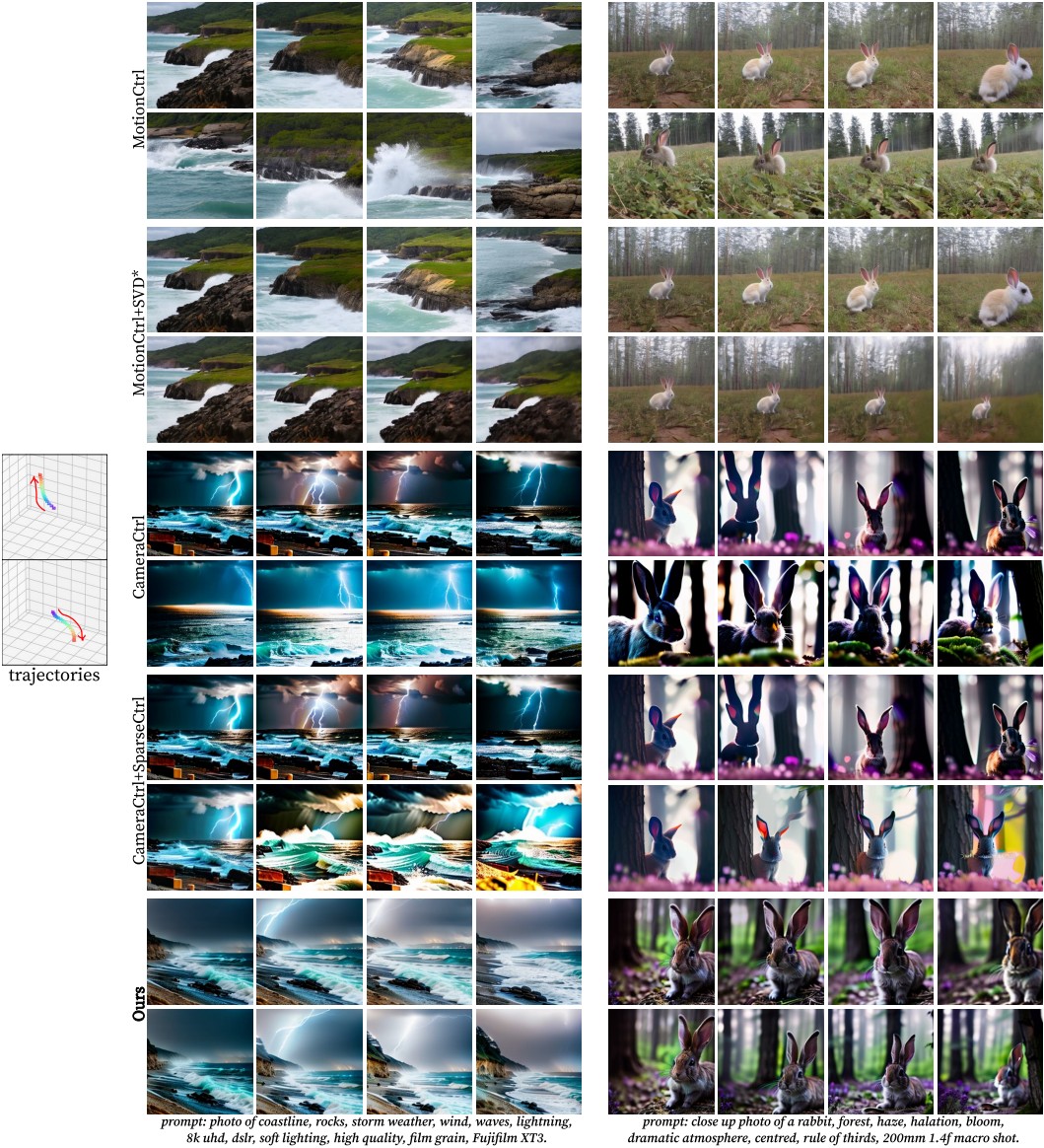

*prompt: photo of coastline, rocks, storm weather, wind, waves, lightning, 8k uhd, dslr, soft lighting, high quality, film grain, Fujifilm XT3.*

*prompt: close up photo of a rabbit, forest, haze, halation, bloom, dramatic atmosphere, centred, rule of thirds, 200mm 1.4f macro shot.*

Figure 4: **Qualitative comparison.** Our method maintains consistency across videos for static and dynamic scenes, while no prior work can generate synchronized different realizations. * Despite our best efforts, we are incapable of getting MotionCtrl [60]+SVD [5] to generate any motion beyond the simplest static camera zooming in-and-out. Please refer to our supplemental video for illustration.

model to generate a reference video first, then take its first frame as the input of their image-to-video version (i.e., their combination with SparseCtrl [16] and SVD [5]) to derive the second video. We use the same metrics as in the first experiment but instead evaluate between the corresponding frames from the two videos. Results are shown in Tab. 1, where our model greatly outperforms all baselines.

**Semantic and fidelity evaluations.** Following the standard practice of prior works [17, 61, 29, 9, 10, 8], we report CLIP [41] embedding similarity between **1)** each frame of the output video and the corresponding input prompt and **2)** pairs of frames across video pairs. The former metric, denoted as CLIP-T, is to show that our model does not destroy the appearance/content prior of our base model, and the latter, denoted as CLIP-F, is aimed to show that the cross-view module can improve the semantic and structural consistency between the generated video pair. For these purposes, we randomly sample 1000 videos using camera trajectory pairs from RealEstate10K, along with text

captions from WebVid10M (2000 videos generated in total). To further demonstrate our method's ability to maintain high-fidelity generation contents, we report FID [20] and KID [4] ×100 using the implementation [38], and FVD [57]. We do not compare against models that do not share the same base model as us for FID [20], KID [4] and FVD [57], since these metrics are strongly influenced by the abilities of the base models. Following prior work [18], we evaluate these two metrics on RealEstate10k [68] because of the strong undesired bias, e.g., watermarks, on WebVid10M [1]. As shown in Tab. 2, our model surpasses all baselines for the CLIP [41]-based metrics. This proves our model's ability to synthesize collaborative videos that share a scene while maintaining and improving fidelity according to the prompt. Our model also performs better than or on par with all prior works on fidelity metrics, which indicates robustness to the appearances and content priors learned by our base models.

## 5.3 Ablation Study

Table 3: **Ablation Study** conducted on generic scenes (prompts from WebVid10M [1]), where we deactivate each of our introduced modules. Results indicate that our full pipeline outperforms the ablation settings for both geometric and semantic consistencies.

| | Rot. AUC (@5°/10°/20°) | Trans. AUC (@5°/10°/20°) | Semantic Consistency | |
|---|---|---|---|---|
| | | | CLIP-T ↑ | CLIP-F ↑ |
| Ours w/o Epi | 16.8 / 31.8 / 49.1 | 1.5 / 5.4 / 13.7 | 0.30 | 0.91 |
| Ours RE10K only | 17.9 / 29.8 / 43.3 | 1.7 / 5.3 / 13.2 | 0.29 | 0.90 |
| Ours w/o HG | 22.0 / 35.5 / 50.5 | 2.3 / 6.1 / 14.5 | 0.29 | 0.92 |
| Ours 1 Layer | 22.7 / 37.8 / 54.3 | 3.1 / 8.5 / 19.2 | 0.29 | 0.92 |
| Ours | **25.2 / 40.7 / 57.5** | **3.7 / 9.6 / 19.9** | **0.30** | **0.93** |

We perform a thorough ablation study in Tab. 3 to verify our design choices, where the variants are: **1)** No epipolar line constraints (*Ours w/o Epi*), where we perform a normal self-attention instead of epipolar attention in our *Cross-View Synchronization Module*; **2)** No mixed training (*Ours RE10K only*), where we follow the setups in CameraCtrl [18] and train the model only on RealEstate10k [68]; **3)** No homography augmentation (*Ours w/o HG*), where we switch off the homography transformations applied to WebVid10M [1] videos during training; and **4)** using only 1 *Cross-View Synchronization Module* instead of 2 (*Ours 1 Layer*). The ablation study indicates that while we can get semantically consistent outputs without epipolar constraints, they are essential to gain geometrical consistency. We also observe that the mixed training strategy and homography augmentation greatly improve all metrics, including semantic consistency, further verifying their purpose of closing the gap between static training scenes and desired dynamic outputs. We believe there are two reasons why our full model outperforms the model trained on RealEstate10K [68]. The first reason is our epipolar attention design. In the WebVid10M [1] training stage, while there are no camera poses available, we use pseudo-gt epipolar lines (i.e. lines calculated from homography matrix H. The line of pixel x in the warped frame goes through the pixel Hx) to describe the spatial relationship between video frames. This enhances the model's ability to generate videos that satisfy the given line conditions. Hence, in a camera-control setting, the full model is more constrained to the epipolar lines and generates videos that align better with the camera poses. Secondly, since RealEstate10K [68] mostly consists of static indoor scenes, models trained on RealEstate10K [68] may suffer from data bias and may not perform well on general scenes, thus resulting in poor evaluation performance in this experiment.

## 6 Discussion

We introduce CVD, a novel framework facilitating collaborative video generation. It ensures seamless information exchange between video instances, synchronizing content and dynamics. Additionally, CVD offers camera customization for comprehensive scene capture with multiple cameras. The core innovation of CVD is its utilization of epipolar geometry, derived from reconstruction pipelines, as a constraint. This geometric framework fine-tunes a pre-trained video diffusion model. The training process is enhanced by integrating dynamic, single-view, in-the-wild videos to maintain a diverse range of motion patterns. During inference, CVD employs a multi-view sampling strategy to facilitate efficient information sharing across videos, resulting in a "collaborative diffusion" effect for unified video output. To our knowledge, CVD represents the first approach to tackle the complexities of

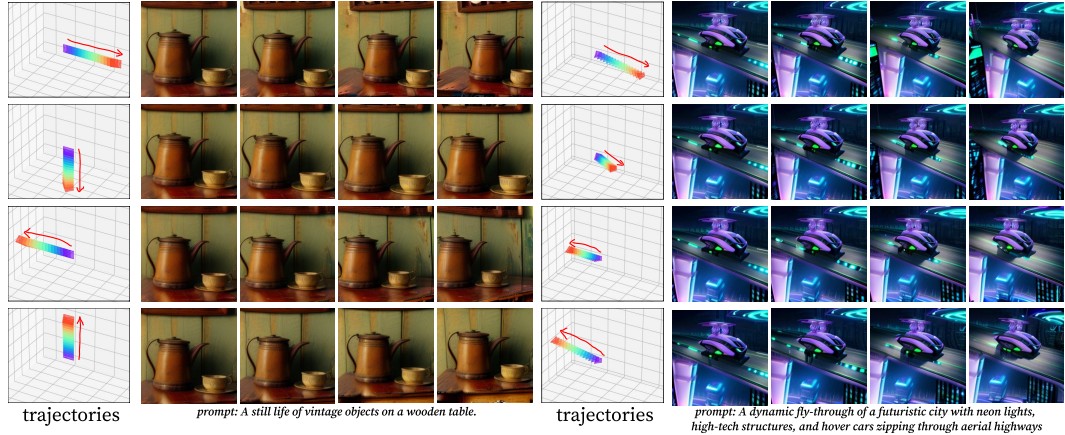

trajectories    *prompt: A still life of vintage objects on a wooden table.*    trajectories    *prompt: A dynamic fly-through of a futuristic city with neon lights, high-tech structures, and hover cars zipping through aerial highways*

Figure 5: **Multi-view Video Generation.** *Left*: The cameras move towards 4 directions, while all cameras are looking at the same 3D point; *Right*: The trajectories are interpolated from one trajectory (*1st Row*) to another (*4th Row*).

multi-view or multi-trajectory video synthesis. It significantly advances beyond existing multi-view image generation technologies, such as Zero123 [35], by also ensuring consistent dynamics across all videos produced. This breakthrough marks a critical development in video synthesis, promising new capabilities and applications.

## 6.1   Limitations

CVD faces certain limitations. Primarily, the effectiveness of CVD is inherently linked to the performance of its base models, AnimateDiff [17] and CameraCtrl [18]. While CVD strives to facilitate robust information exchange across videos, it does not inherently solve the challenge of internal consistency within individual videos. As a result, issues such as uncanny shape shifting and dynamic inconsistencies that are presented in the base models may persist, affecting the overall consistency across the video outputs. Additionally, it cannot synthesize videos in real time, owing to the computationally intensive nature of diffusion models. Nevertheless, the field of diffusion model optimization is rapidly evolving, and forthcoming advancements are likely to enhance the efficiency of CVD significantly.

## 6.2   Broader Impacts

Our approach represents a significant advancement in multi-camera video synthesis, with wide-ranging implications for industries such as filmmaking and content creation. However, we are mindful of the potential misuse, particularly in creating deceptive content like deepfakes. We categorically oppose the exploitation of our methodology for any purposes that infringe upon ethical standards or privacy rights. To counteract the risks associated with such misuse, we advocate for the continuous development and improvement of deepfake detection technologies.

**Acknowledgement**   This project was partly supported by Google and Samsung.

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

# A Appendix / supplemental material

## A.1 More Analysis on Collaborative Video Generation.

For simplicity, the video conditions are omitted in this section without loss of generality. In the paper, we describe the collaborative video diffusion model as a multivariate denoising function $p_\theta(v_t^{1,...,M})$ that estimates the real distribution $q(v_{t-1}^{1,...,M}|v_t^{1,...,M}, v_0^{1,...,M})$. Following Ho et.al. [22], the problem can be transformed into the optimizing a noise prediction network $\epsilon_\theta(v_t^{1,...,M})$ to predict $\epsilon_t = \frac{1}{\sqrt{1-\bar{\alpha}_t}}v_t^{1,...,M} - \frac{\sqrt{\bar{\alpha}_t}}{\sqrt{1-\bar{\alpha}_t}}v_0^{1,...,M}$. On the other hand, Song et.al. [52] demonstrated the relation between noise prediction and the score function $s_\theta(v_t^{1,...,M}, t) \approx \nabla \log q(v_t^{1,...,M})$ is:

$$s_\theta(v_t^{1,...,M}, t) = -\frac{\epsilon_\theta(v_t^{1,...,M}, t)}{\sqrt{1-\bar{\alpha}_t}}. \tag{5}$$

As discussed in the paper, directly training $s_\theta(v_t^{1,...,M}, t)$ for an arbitrary $M$ is intractable due to the lack of multi-view datasets, so we reduce the problem into generating video pairs ($M = 2$) instead. Specifically, we train a score function $s_\theta(v^{i,j})$ using video pair datasets and apply it to infer all $M$ videos. Our collaborative video score function, denoted as $s_{\text{CVD}}(v^{1,...,M})$, is defined as:

$$s_{\text{CVD}}(v^{1,...,M}) \doteq \sum_{(i,j)\in\mathcal{P}} w_i^{i,j} s_\theta(v^{i,j})_i + w_j^{i,j} s_\theta(v^{i,j})_j, \tag{6}$$

where $\mathcal{P}$ is the set of all selected video pairs, and $s_\theta(v^{i,j})_i = e_i s_\theta(v^{i,j})$ represents the $i$'th video component of the score function. Note that the order of $i, j$ is irrelevant. In essence, we utilize the weighted sum of video pair score functions to depict the score function of all videos. We demonstrate that the defined score function $s_{\text{CVD}}(v^{1,...,M})$ can estimate the real score function $\nabla \log q(v^{1,...,M})$, only if $\sum_{(i,j)\in\mathcal{P}} w_i^{i,j} = 1$ for all $i \in 1, ..., M$.

**Lemma A.1.** *Let $S$ be a subset of $\{1, ..., M\}$, and $q(v_t^S)$ be the density function of a set of video features $v_t^S = \{v_t^k | k \in S\}$ derived from the forward diffusion process, that is, $q(v_t^S \mid v_0^S) = \mathcal{N}(v_t^S; \sqrt{\bar{\alpha}_t}v_0^S, (1-\bar{\alpha}_t)I)$. Then $\nabla_{v_t^k} \log q(v_t^S | v_0^S) = \frac{\mathbf{1}(k\in S)}{1-\bar{\alpha}_t}(\sqrt{\bar{\alpha}_t}v_0^k - v_t^k)$, where $\mathbf{1}(k \in S)$ equals to 1 if $k \in S$ and 0 otherwise.*

*Proof.*

$$\nabla_{v_t^k} \log q(v_t^S|v_0^S) = \nabla_{v_t^k} \log(\mathcal{N}(v_t^S; \sqrt{\bar{\alpha}_t}v_0^S, (1-\bar{\alpha}_t)I)) \tag{7}$$

$$= \nabla_{v_t^k} \frac{-(v_t^S - \sqrt{\bar{\alpha}_t}v_0^S)^2}{2(1-\bar{\alpha}_t)} \tag{8}$$

$$= \frac{\mathbf{1}(k \in S)}{1-\bar{\alpha}_t}(\sqrt{\bar{\alpha}_t}v_0^k - v_t^k) \tag{9}$$

$\square$

**Lemma A.2** (Updated Tweedies's Formula). *Let $S$ be a subset of $\{1, ..., M\}$, and $q(v_t^S)$ be the density function of a set of video features $v_t^S = \{v_t^k | k \in S\}$ derived from the forward diffusion process, then $\nabla_{v_t^k} \log q(v_t^S) = \frac{\mathbf{1}(k\in S)}{1-\bar{\alpha}_t}(\sqrt{\bar{\alpha}_t}E_q(v_0^k|v_t^S) - v_t^k)$.*

*Proof.*

$$\nabla_{\boldsymbol{v}_t^k} \log q(\boldsymbol{v}_t^S) = \frac{\nabla_{\boldsymbol{v}_t^k} q(\boldsymbol{v}_t^S)}{q(\boldsymbol{v}_t^S)} \tag{10}$$

$$= \frac{\nabla_{\boldsymbol{v}_t^k} \boldsymbol{E}_{\boldsymbol{v}_0^S}(q(\boldsymbol{v}_t^S|\boldsymbol{v}_0^S))}{q(\boldsymbol{v}_t^S)} \tag{11}$$

$$= \frac{\boldsymbol{E}_{\boldsymbol{v}_0^S}(\nabla_{\boldsymbol{v}_t^k} q(\boldsymbol{v}_t^S|\boldsymbol{v}_0^S))}{q(\boldsymbol{v}_t^S)} \tag{12}$$

$$= \int \frac{q(\boldsymbol{v}_0^S)}{q(\boldsymbol{v}_t^S)} \nabla_{\boldsymbol{v}_t^k} q(\boldsymbol{v}_t^S|\boldsymbol{v}_0^S) \, d\boldsymbol{v}_0^S \tag{13}$$

$$= \int q(\boldsymbol{v}_0^S|\boldsymbol{v}_t^S) \nabla_{\boldsymbol{v}_t^k} \log q(\boldsymbol{v}_t^S|\boldsymbol{v}_0^S) \, d\boldsymbol{v}_0^S \quad \text{(Bayes' Theorem)} \tag{14}$$

$$= \int q(\boldsymbol{v}_0^S|\boldsymbol{v}_t^S) \cdot \frac{\mathbf{1}(k \in S)}{1 - \bar{\alpha}_t} (\sqrt{\bar{\alpha}_t} \boldsymbol{v}_0^k - \boldsymbol{v}_t^k) \, d\boldsymbol{v}_0^S \quad \text{(Lemma. A.1)} \tag{15}$$

$$= \frac{\mathbf{1}(k \in S)}{1 - \bar{\alpha}_t} (\sqrt{\bar{\alpha}_t} \int q(\boldsymbol{v}_0^S|\boldsymbol{v}_t^S) \boldsymbol{v}_0^k \, d\boldsymbol{v}_0^S - \boldsymbol{v}_t^k) \tag{16}$$

$$= \frac{\mathbf{1}(k \in S)}{1 - \bar{\alpha}_t} (\sqrt{\bar{\alpha}_t} \int q(\mathbf{v}_0^S|\mathbf{v}_t^S) \mathbf{v}_0^k \, d\mathbf{v}_0^S - \mathbf{v}_t^k) \tag{17}$$

$$= \frac{\mathbf{1}(k \in S)}{1 - \bar{\alpha}_t} (\sqrt{\bar{\alpha}_t} \int q(\mathbf{v}_0^k|\mathbf{v}_t^S) q(\mathbf{v}_0^{S/k}|\mathbf{v}_0^k, \mathbf{v}_t^S) \mathbf{v}_0^k \, d\mathbf{v}_0^S - \mathbf{v}_t^k) \tag{18}$$

$$= \frac{\mathbf{1}(k \in S)}{1 - \bar{\alpha}_t} (\sqrt{\bar{\alpha}_t} \int q(\mathbf{v}_0^k|\mathbf{v}_t^S) \mathbf{v}_0^k \int q(\mathbf{v}_0^{S/k}|\mathbf{v}_0^k, \mathbf{v}_t^S) \, d\mathbf{v}_0^{S/k} \, d\mathbf{v}_0^k - \mathbf{v}_t^k) \tag{19}$$

$$= \frac{\mathbf{1}(k \in S)}{1 - \bar{\alpha}_t} (\sqrt{\bar{\alpha}_t} \int q(\mathbf{v}_0^k|\mathbf{v}_t^S) \mathbf{v}_0^k \, d\mathbf{v}_0^k - \mathbf{v}_t^k) \tag{20}$$

$$= \frac{\mathbf{1}(k \in S)}{1 - \bar{\alpha}_t} (\sqrt{\bar{\alpha}_t} \int q(\boldsymbol{v}_0^k|\boldsymbol{v}_t^S) q(\boldsymbol{v}_0^{S/k}|\boldsymbol{v}_0^k, \boldsymbol{v}_t^S) \boldsymbol{v}_0^k \, d\boldsymbol{v}_0^S - \boldsymbol{v}_t^k) \tag{21}$$

$$= \frac{\mathbf{1}(k \in S)}{1 - \bar{\alpha}_t} (\sqrt{\bar{\alpha}_t} \int q(\boldsymbol{v}_0^k|\boldsymbol{v}_t^S) \boldsymbol{v}_0^k \int q(\boldsymbol{v}_0^{S/k}|\boldsymbol{v}_0^k, \boldsymbol{v}_t^S) \, d\boldsymbol{v}_0^{S/k} \, d\boldsymbol{v}_0^k - \boldsymbol{v}_t^k) \tag{22}$$

$$= \frac{\mathbf{1}(k \in S)}{1 - \bar{\alpha}_t} (\sqrt{\bar{\alpha}_t} \int q(\boldsymbol{v}_0^k|\boldsymbol{v}_t^S) \boldsymbol{v}_0^k \, d\boldsymbol{v}_0^k - \boldsymbol{v}_t^k) \tag{23}$$

$$= \frac{\mathbf{1}(k \in S)}{1 - \bar{\alpha}_t} (\sqrt{\bar{\alpha}_t} \boldsymbol{E}_q(\boldsymbol{v}_0^k|\boldsymbol{v}_t^S) - \boldsymbol{v}_t^k) \tag{24}$$

$\square$

**Theorem A.3.** *The function $\boldsymbol{s}_{CVD}(\boldsymbol{v}_t^{1,...,M})$ can be an unbiased approximation of the real score function $\nabla \log q(\boldsymbol{v}_t^{1,...,M})$ for all timesteps $t \in 1, ..., T$, only if $\sum_{(i,j)\in\mathcal{P}} w_i^{i,j} = 1$ for all $i \in 1, ..., M$.*

*Proof.* For any $k \in 1, .., M$, the $k$'th component of $\boldsymbol{s}_{CVD}(\boldsymbol{v}_t^{1,...,M})$ can be written as:

$$\boldsymbol{s}_{\text{CVD}}(\boldsymbol{v}_t^{1,...,M})_k \tag{25}$$

$$= (\sum_{(i,j)\in\mathcal{P}} w_i^{i,j} \boldsymbol{s}_\theta(\boldsymbol{v}^{i,j})_i + w_j^{i,j} \boldsymbol{s}_\theta(\boldsymbol{v}^{i,j})_j)_k \tag{26}$$

$$= \sum_{(k,j)\in\mathcal{P}} w_k^{k,j} \boldsymbol{s}_\theta(\boldsymbol{v}^{k,j})_k \tag{27}$$

$$\approx \sum_{(k,j)\in\mathcal{P}} w_k^{k,j} \nabla_{\boldsymbol{v}^k} \log q(\boldsymbol{v}^{k,j}) \quad \text{(Score Matching)} \tag{28}$$

$$= \frac{1}{1 - \bar{\alpha}_t} \sum_{(k,j)\in\mathcal{P}} w_k^{k,j} (\sqrt{\bar{\alpha}_t} \boldsymbol{E}_q(\boldsymbol{v}_0^k|\boldsymbol{v}_t^{k,j}) - \boldsymbol{v}_t^k) \quad \text{(Lemma. A.2)} \tag{29}$$

To unbiasedly estimate $\nabla_{\boldsymbol{x}_t^k} \log q(\boldsymbol{v}_t^{1,...,M}) = \frac{1}{1-\bar{\alpha}_t}(\sqrt{\bar{\alpha}_t}\boldsymbol{E}_q(\boldsymbol{v}_0^k|\boldsymbol{v}_t^{1,...,M}) - \boldsymbol{v}_t^k)$ from Eq. 29 w.r.t. all $t$ and $\boldsymbol{v}_t^k$, there must be $\sum_{(k,j)\in\mathcal{P}} w_k^{k,j}\boldsymbol{v}_t^k = \boldsymbol{v}_t^k$, which means $\sum_{(k,j)\in\mathcal{P}} w_k^{k,j} = 1$. $\qquad\square$

In addition, we can observe that the accuracy of the estimation from Eq. 29 heavily relies on the similarity between $\sum_{(k,j)\in\mathcal{P}} w_k^{k,j}\boldsymbol{E}_q(\boldsymbol{v}_0^k|\boldsymbol{v}_t^{k,j})$ and $\boldsymbol{E}_q(\boldsymbol{v}_0^k|\boldsymbol{v}_t^{1,...,M})$. That means, when we apply a denoising step to a noisy input $\boldsymbol{v}_t^{1,...,M}$, the output $\boldsymbol{v}_{t-1}^{1,...,M}$ is more likely to align with the true distribution if the prediction of $\boldsymbol{v}_0^k$ from $\boldsymbol{v}_t^{k,j}$ resembles the prediction of $\boldsymbol{v}_0^k$ from all $\boldsymbol{v}_t^{1,...,M}$. We think this is fairly reasonable in the context of consistent camera-controlled video generation, as the underlying geometry of captured videos can often be discerned from just a few views. We believe this is the key reason why our model can generate consistent multi-view videos trained from video pair data only.

## A.2 Implementing Details

We built our pipeline on top of AnimateDiff [17], a popular open-source T2V model that is widely used among artists. We additionally deploy CameraCtrl [18] to utilize its camera conditioning ability. Following this line of works, we benefit from their plug-and-play property and can swap our base model with a fine-tuned version, e.g., via DreamBooth [45] or LORA [25].

### A.2.1 Training

We select 65,000 videos from RealEstate10K [68] and 2,400,000 videos from WebVid10M [1] to train our model. Each data point consists of two videos of 16 frames and their corresponding camera extrinsic and intrinsic parameters. For RealEstate10K, we randomly sample a 31-frame clip from the original video and split it into two videos using the method described in the paper. For WebVid10M, we sample a 16-frame clip, duplicate it to create two videos, and then apply random homography deformations to the second video. The homography transformation matrix $H = H_t H_r H_s H_{sh} H_p$ is defined as the composition of a series of transformations: translation, rotation, scaling, shearing, and projection, where:

$$H_t = \begin{bmatrix} 1 & 0 & t_0 \\ 0 & 1 & t_1 \\ 0 & 0 & 1 \end{bmatrix}, H_r = \begin{bmatrix} \cos(\theta) & -\sin(\theta) & 0 \\ \sin(\theta) & \cos(\theta) & 0 \\ 0 & 0 & 1 \end{bmatrix},$$

$$H_s = \begin{bmatrix} 1+s_0 & 0 & 0 \\ 0 & 1+s_1 & 0 \\ 0 & 0 & 1 \end{bmatrix}, H_{sh} = \begin{bmatrix} 1 & sh_0 & 0 \\ sh_1 & 1 & 0 \\ 0 & 0 & 1 \end{bmatrix}, H_p = \begin{bmatrix} 1 & 0 & 0 \\ 0 & 1 & 0 \\ p_0 & p_1 & 1 \end{bmatrix} \tag{30}$$

are transformation matrices parameterized by controlling vectors $t, \theta, s, sh, p$. We aim for the first frame of the deformed video to remain unchanged, with the deformation gradually increasing in subsequent frames. To achieve this, we randomly sample the controlling vectors for the last frame from normal distributions. Then, we interpolate these vectors from 0 to the sampled values to obtain the vectors for each intermediate frame and calculate the corresponding matrices.

Following [18], we use the Adam optimizer [30] with learning rate $1e-4$. During training, we freeze the vanilla parameters from our backbones and optimize only our newly injected layers. We mix the data points from RealEstate10K and WebVid10M under the ratio of $7:3$ and train the model in two phases alternatively. All models are trained on 8 NVIDIA A100 GPUs for 100k iterations using an effective batch size 8. The training takes approximately 30 hours.

### A.2.2 Inference

We use DDIM [51] scheduler with 1000 steps during training and 25 steps during inference. Assuming $w_k^{k,j}$ is independent with $j$, we have $w_k^{k,j} = w_k = \frac{1}{|(k,j)\in\mathcal{P}|}$. Our algorithm is shown in Alg. 1. We use the partitioning strategy in all of our experiments. For 2-view (video pair) results, we use $R=1$(no recurrent denoising) and $P=1$; For 4-view results, we use $R=4, P=1$; and for 6-view results, we use $R=6, P=2$. We demonstrate multi-view video generation results in our supplementary videos. Additionally, we show three potential applications of our algorithm: long looping videos, view switching, and potential 3D generation.

**Algorithm 1:** Algorithm for arbitrary number of videos generation

---

**Parameter:** Denoising steps $T$, recurrent steps $R$, video number $M$, noise scheduling parameters $\{\bar{\alpha}_t\}_{t=1}^{T}$, pair selecting strategy $Stg \in \{$Exhaustive, Partition$\}$, partition number $Q$

**Input:** $\boldsymbol{v}_T^{1,...,M}$ sampled from $\mathcal{N}(0, I)$, video pair diffusion model $\epsilon_\theta$, text prompt $y$, camera trajectories $cam^{1,...,M}$

**for** $t = T, T-1, ..., 1$ **do**

    $\epsilon_{\text{out}}^{1,...,M} \leftarrow 0$;

    **for** $r = 0, 1, ..., R-1$ **do**

        **if** *Stg is Exhaustive* **then**

            $\mathcal{P} \leftarrow \{(i,j) \mid i,j \in 1, ..., M, i \neq j\}$ ;        `/* Selecting all pairs */`

            $denom \leftarrow M - 1$;

        **else**

            $\mathcal{P} \leftarrow \{\}$;

            **for** $q = 0, 1, ..., Q-1$ **do**

                $\mathcal{P}$.Extend(RandomPairPartition$(1, 2, .., M)$)

            **end**

            $denom \leftarrow Q$;

        **end**

        **for** $(i,j) \in \mathcal{P}$ **do**

            $\epsilon_{\text{out}}^i \leftarrow \epsilon_{\text{out}}^i + \boldsymbol{\epsilon}_\theta^i(\boldsymbol{v}_t^{i,j}, t, cam^{i,j})$;

            $\epsilon_{\text{out}}^j \leftarrow \epsilon_{\text{out}}^j + \boldsymbol{\epsilon}_\theta^j(\boldsymbol{v}_t^{i,j}, t, cam^{i,j})$;

        **end**

        $\boldsymbol{v}_{t-1}^{1,...,M} = \text{NoiseSchedule}(\epsilon_{\text{out}}^{1,...,M}/denom, \boldsymbol{v}_t^{1,...,M}, t)$;

        **if** $r \neq R-1$ **then**

            $\epsilon' \sim \mathcal{N}(0, I)$;

            $\boldsymbol{v}_t^{1,...,M} = \sqrt{\bar{\alpha}_t/\bar{\alpha}_{t-1}}\boldsymbol{v}_{t-1}^{1,...,M} + \sqrt{1 - \bar{\alpha}_t/\bar{\alpha}_{t-1}}\epsilon'$ ;    `/* Renoise */`

        **end**

    **end**

**end**

---

## A.3  Results of Attention Maps

We show an exemplar visualization of our epipolar-based attention in Fig. 6, where we take the highlighted pixel from the left image, and visualize its corresponding attention probability after softmax. We can observe that information is taken from the second image according to the epipolar line, and specifically, the corresponding region is being attended to.

## A.4  Performances with identical camera trajectories

In Fig. 7, we show that our model can generate identical videos if the input camera trajectories are identical, while none of the prior works communicates cross-videos, hence incapable of generating identical contents. Quantitatively, our model reaches an MSE of 0.01, significantly outperforming CameraCtrl at 0.07 and CameraCtrl+SparseCtrl at 0.06. We show more realizations of our model when the camera trajectory pair and prompt are identical in Fig. 8

## A.5  Additional results for LoRA fine-tuned models

Our model exhibits strong plug-and-play properties and can directly generalize to different fine-tuned models, e.g., using Dreambooth [45] or LoRA [25]. We show a few rendering results in Fig. 9. For better illustration, please refer to our supplemental video.

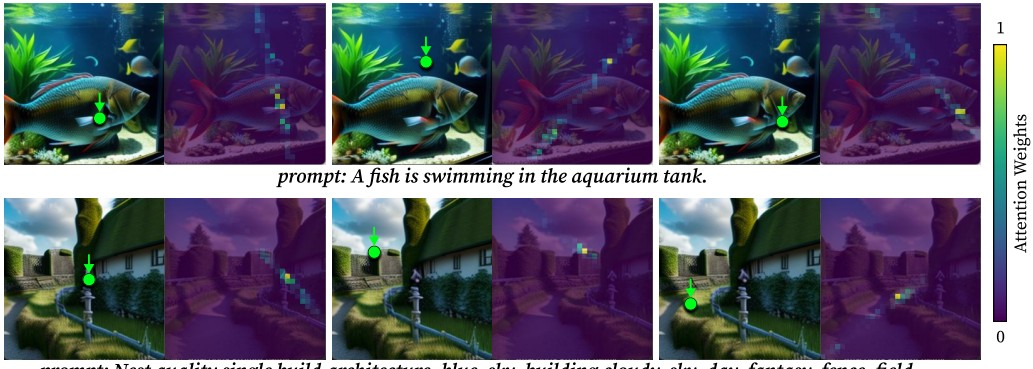

*prompt: A fish is swimming in the aquarium tank.*

*prompt: Nest quality,single build,architecture, blue_sky, building,cloudy_sky, day, fantasy, fence, field, ....*

Figure 6: **Exemplar visualization of epipolar-attention map**.

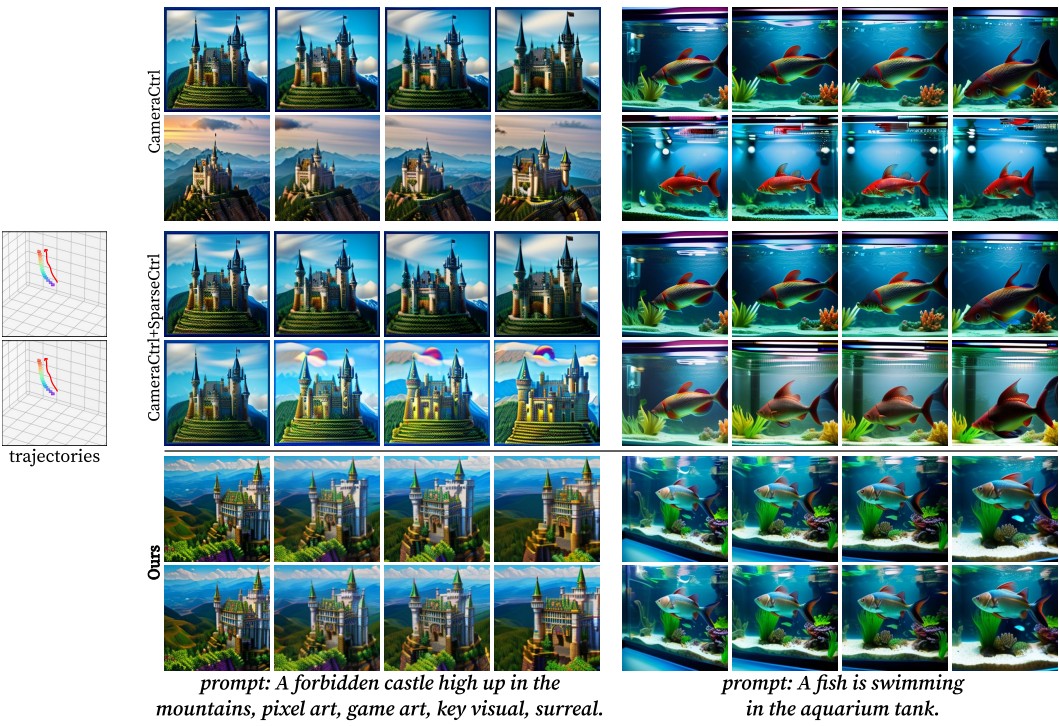

*prompt: A forbidden castle high up in the mountains, pixel art, game art, key visual, surreal.*

*prompt: A fish is swimming in the aquarium tank.*

Figure 7: **Qualitative comparison** with our baselines where the two camera trajectories are identical.

## A.6   More Qualitative Results

Figures  10, 11, 12, 13 and 14 show more qualitative results, where we generate video pairs with different realizations and camera trajectories for each prompt.  Please refer to our supplementary video for better illustrations.

## A.7   Homography warping Visualization

In Fig. 15, we show visualization results of our homography warping augmentation applied in our WebVid10M [1] phase.  During our training, we removed the L2 loss in the potentially unseen pixels (black regions) of the cloned video for data integrity.

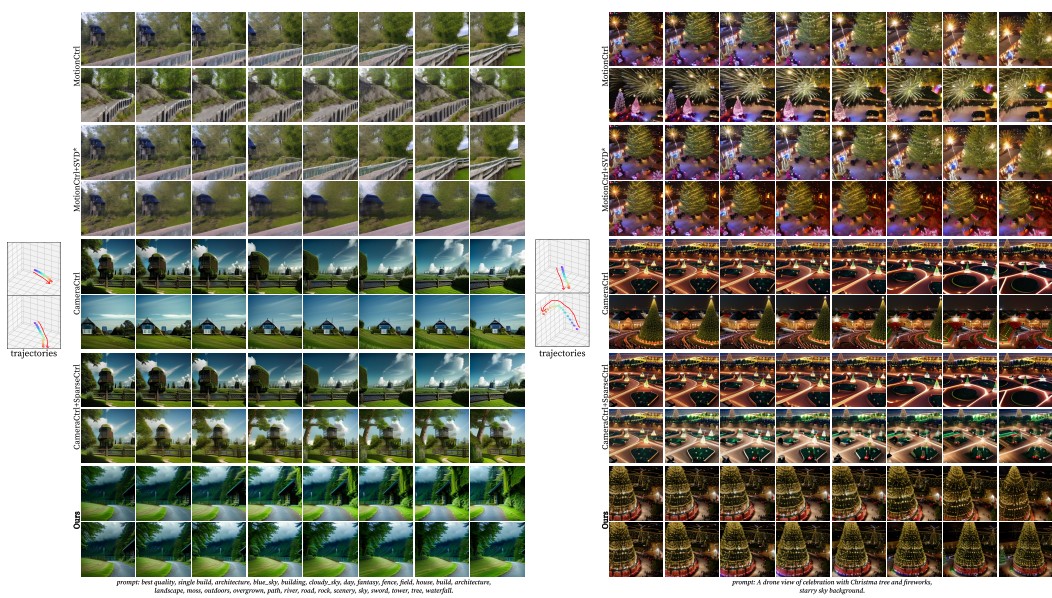

Figure 8: **Qualitative results** where the two camera trajectories are identical.

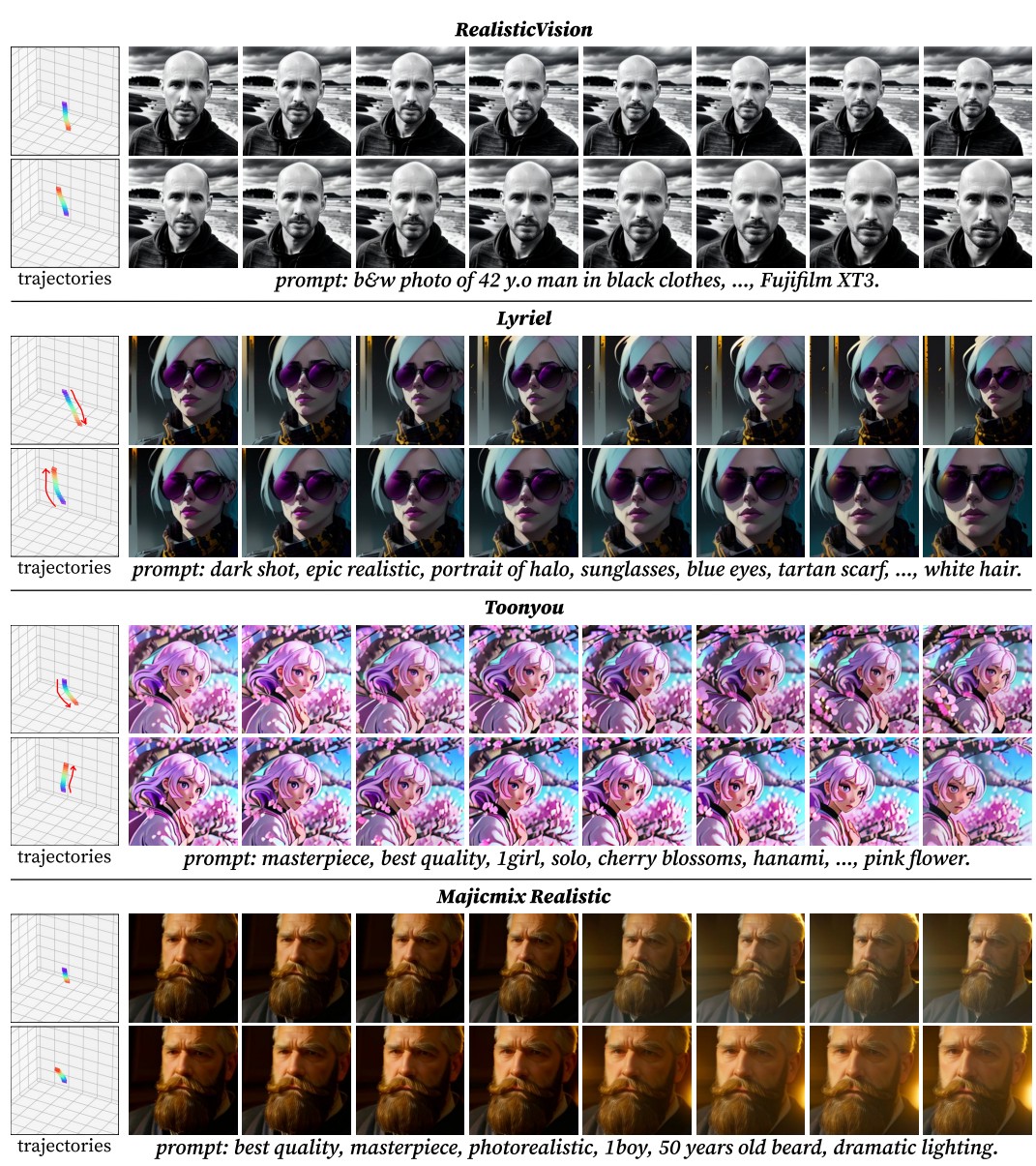

Figure 9: **Exemplar outputs from Dreambooth [45]/LoRA [25] fine-tuned models**.

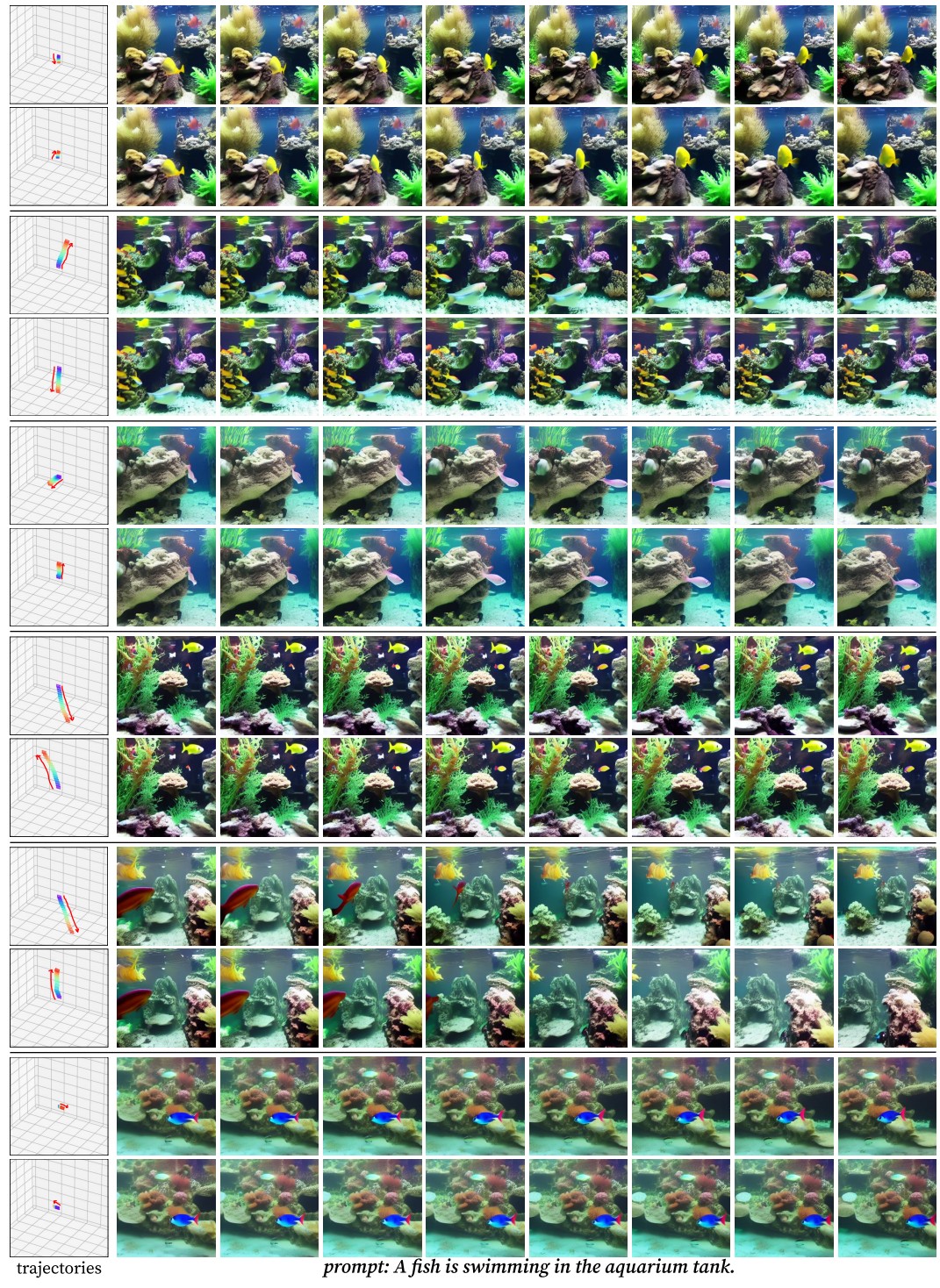

trajectories

*prompt: A fish is swimming in the aquarium tank.*

Figure 10: **Additional Qualitative Results** with different camera trajectories and realizations.

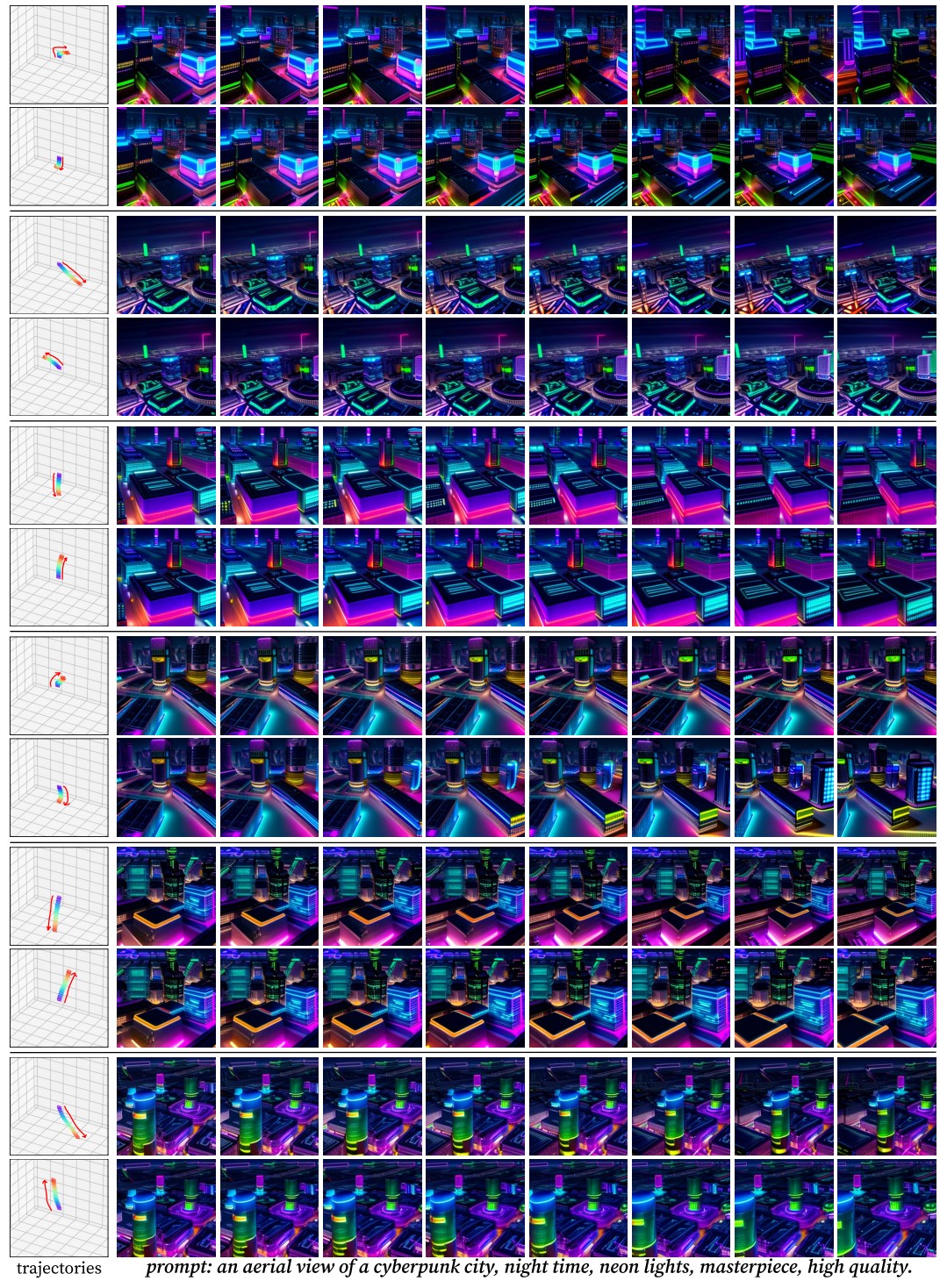

trajectories     *prompt: an aerial view of a cyberpunk city, night time, neon lights, masterpiece, high quality.*

Figure 11: **Additional Qualitative Results** with different camera trajectories and realizations.

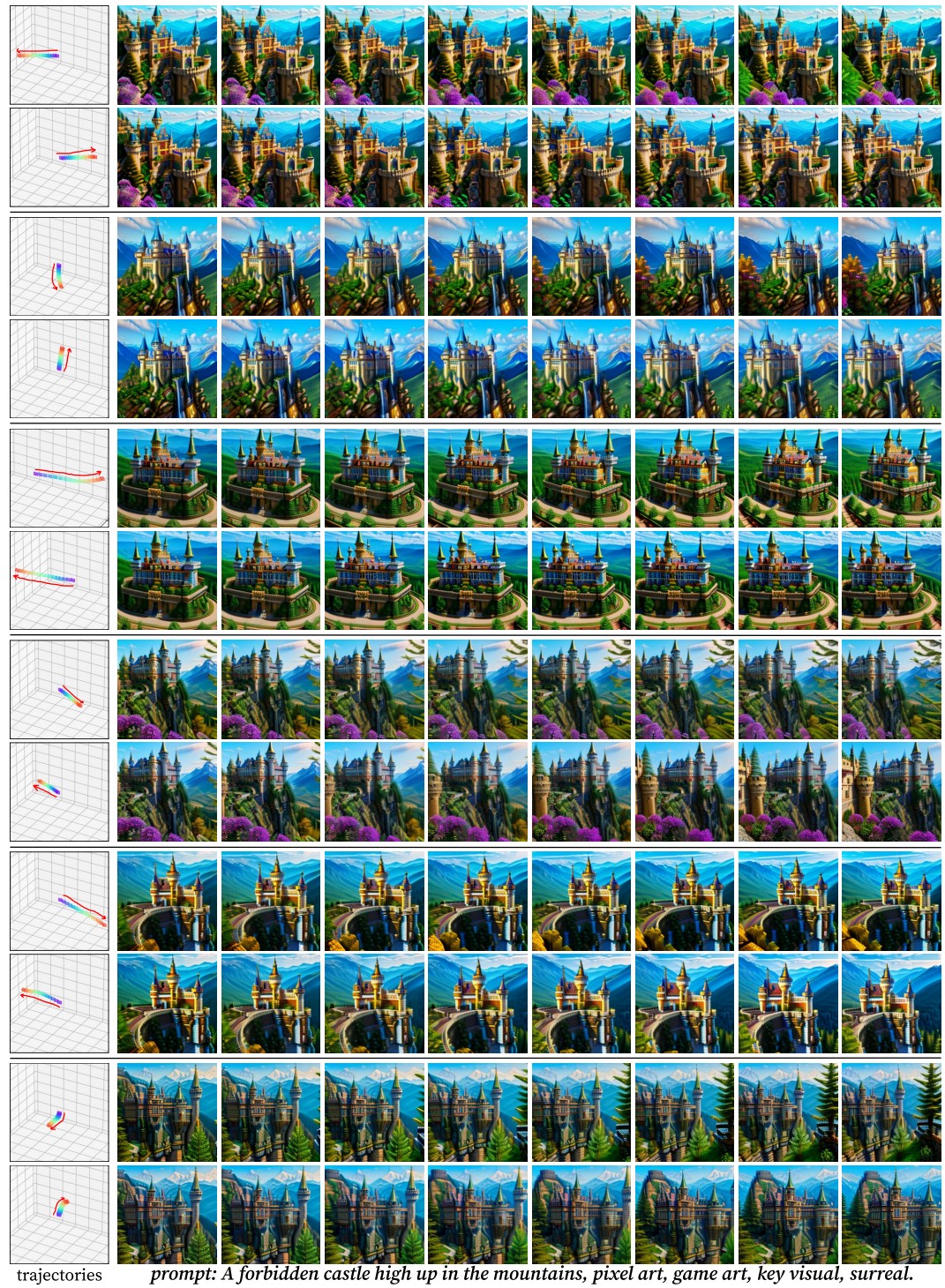

trajectories     *prompt: A forbidden castle high up in the mountains, pixel art, game art, key visual, surreal.*

Figure 12: **Additional Qualitative Results** with different camera trajectories and realizations.

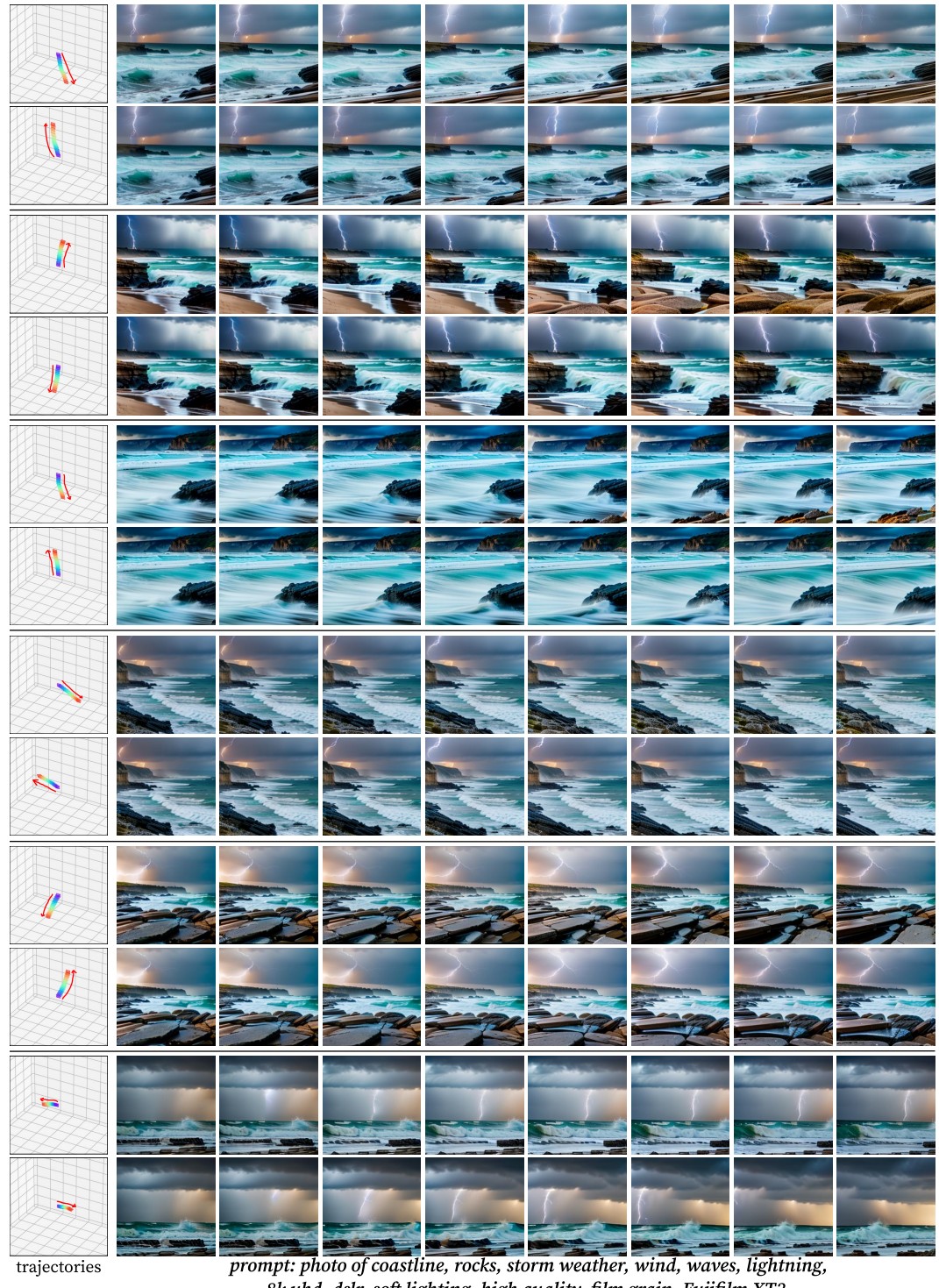

trajectories

*prompt: photo of coastline, rocks, storm weather, wind, waves, lightning, 8k uhd, dslr, soft lighting, high quality, film grain, Fujifilm XT3.*

Figure 13: **Additional Qualitative Results** with different camera trajectories and realizations.

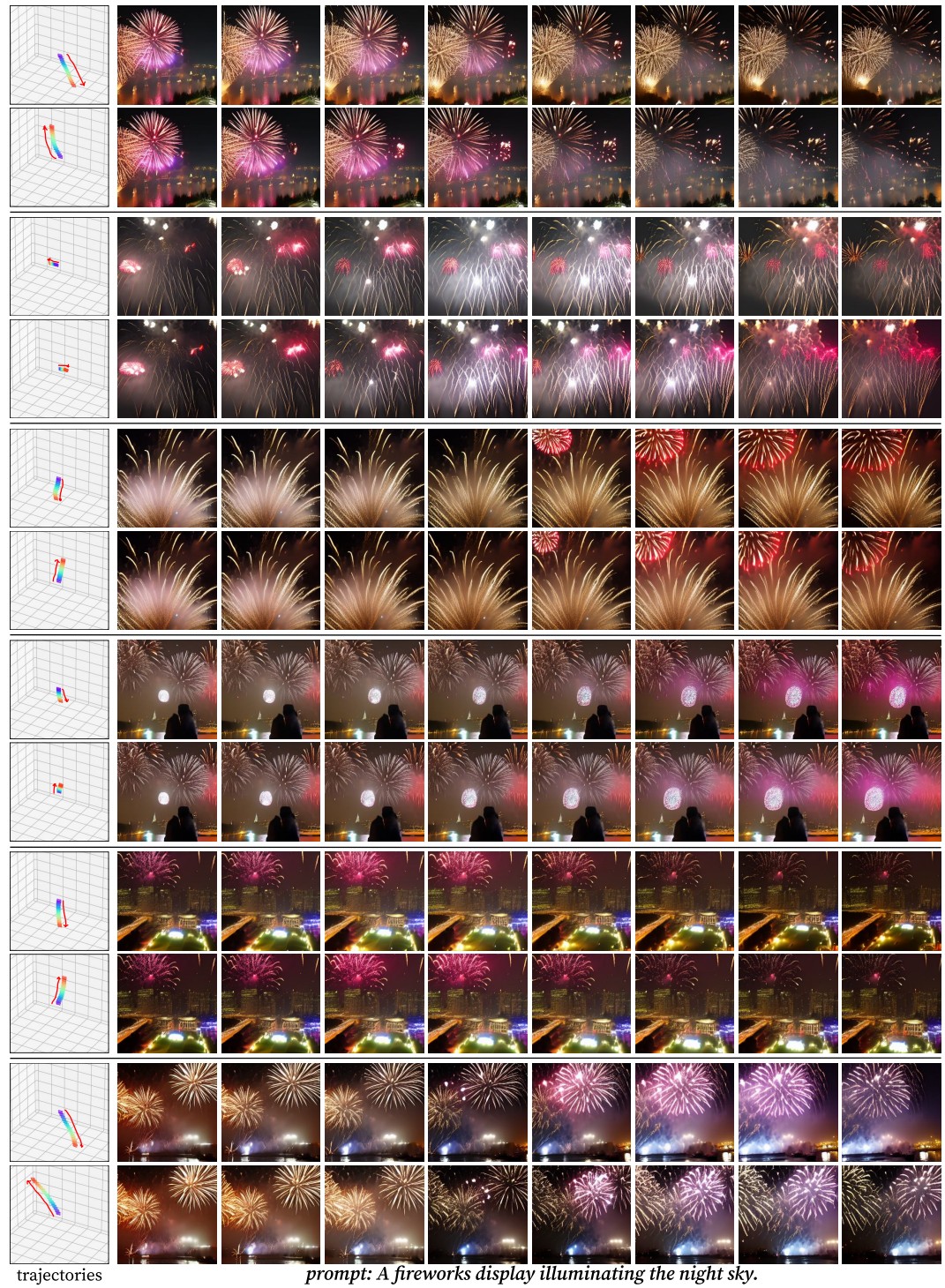

trajectories

*prompt: A fireworks display illuminating the night sky.*

Figure 14: **Additional Qualitative Results** with different camera trajectories and realizations.

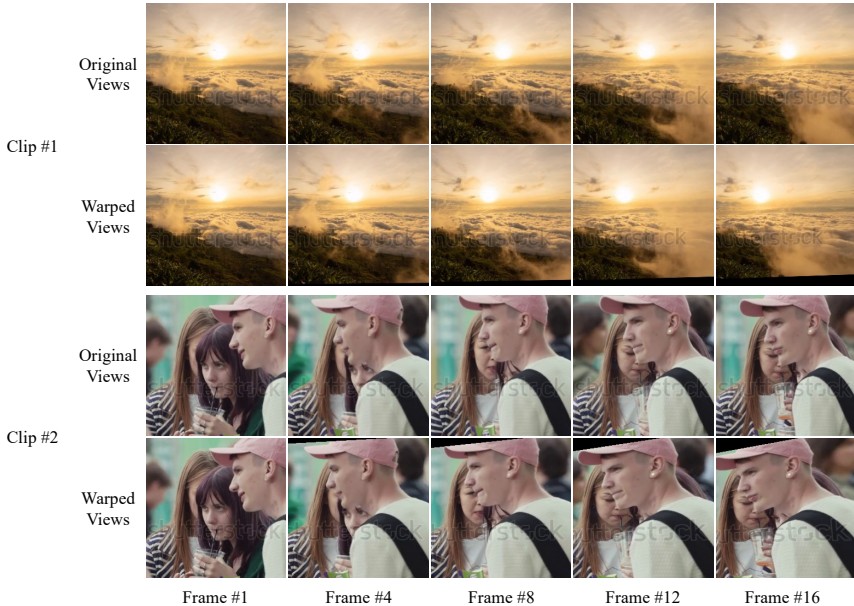

Figure 15: **Visualization of homography warping for WebVid10M data**. For each video clip, the top row represents the original frames from the video, and the bottom row represents the frames warped by homography transformations.

