# OpenReview forum: "Collaborative Video Diffusion: Consistent Multi-video Generation with Camera Control"
_NeurIPS.cc/2024/Conference — NeurIPS 2024 poster_

### Official Review · Reviewer_v3P6 · 2024-07-08

**Soundness:** 3
**Presentation:** 3
**Contribution:** 3
**Rating:** 7
**Confidence:** 4

**Summary:**

This paper tackles the problem of consistent multi-video generation, i.e., generating multiple videos capturing the same scene from various camera trajectories. For this, it proposes a cross-view synchronization module (CVD) based on the epipolar geometry. Training-wise, it proposes a hybrid training strategy, exploiting various video datasets. Quantitative and qualitative experiments demonstrate the effectiveness of the proposed approach for generating consistent videos.

**Strengths:**

- originality-wise: the idea of utilizing epipolar geometry to encourage consistency across generated videos is interesting;
- quality-wise: the provided qualitative results demonstrate high-quality consistency
- clarity-wise: the paper is well-written and easy to follow. The derivation provided in the supplementary is clear.
- significance-wise: the problem of generating consistent video content is vital for enabling fine-grained control for video generation

**Weaknesses:**

## 1. Clarifications about the training strategy

I am confused about how the training strategy in Sec. 4.1 is applied to the trainable parameters, i.e., cross-view synchronization module (CVSM). To be specific, are CVSM blocks shown in Fig. 3 the same model? Namely, will the CVSM in Fig. 3.(b) initialize from the trained CVSM in Fig. 3.(a)?

## 2. Problems in Derivation

I think there are issues from Eq. (16) to (17) in Sec. A of the supplementary. Specifically, this step directly changes $q(v_0^S \vert v_t^S)$ to $q(v_0^k \vert v_t^S)$, which is not suitable.

The main concern I have is that $q(v_0^S \vert v_t^S) \neq \prod_{k\in S} q(v_0^k \vert v_t^S)$ since the videos $\\{ v_0^k \\}_{k \in S}$ captures the same scene and are not independent.

Please correct or clarify.

## 3. Missing related works

Recently, there have been several works tackling the problem of generating multi-view consistent content via manipulating the latent space, e..g, [a, b]. Please provide some discussions for comparison to such line of work.

[a] Kapon et al., MAS: Multi-view Ancestral Sampling for 3D motion generation using 2D diffusion. CVPR 2024
[b] Jiang et al., MVHuman: Tailoring 2D Diffusion with Multi-view Sampling For Realistic 3D
Human Generation. ArXiv 2023

**Questions:**

See "weakness".

**Limitations:**

Yes.

---

> ### Author Rebuttal · Authors · 2024-08-07
>
> # Response of Reviewer v3P6’s Review
>
> ## Q1: Clarification of training Strategy.
>
> The two training phases are applied to the same CVSM parameters in a single training pass. (The CVSM in Fig.3a and Fig.3b are the same modules.) Specifically, we blend the data from Webvid10M or RealEstate10K together. Then in each training iteration, we randomly pick one data batch from our hybrid dataset and train our model in the corresponding phase.
>
> ## Q2: Problem in Derivation from Eq. (16) to Eq. (17):
>
> The transformation from Eq. (16) to Eq. (17) can be expanded as:
>
> $$ \frac{\textbf{1}(k\in S)}{1-\bar\alpha_t} ( \sqrt{\bar \alpha_t} \int q(\textbf{v}_0^S | \textbf{v}_t^S) \textbf{v}_0^k \ d\textbf{v}^S_0 - \textbf{v}_t^k) $$
> $$ = \frac{\textbf{1}(k\in S)}{1-\bar\alpha_t} ( \sqrt{\bar \alpha_t} \int q(\textbf{v}_0^k | \textbf{v}_t^S) q(\textbf{v}_0^{S/k} | \textbf{v}_0^{k}, \textbf{v}_t^S) \textbf{v}_0^k \ d\textbf{v}^S_0 - \textbf{v}_t^k) $$
> $$ =  \frac{\textbf{1}(k\in S)}{1-\bar\alpha_t} ( \sqrt{\bar \alpha_t} \int q(\textbf{v}_0^k | \textbf{v}_t^S) \textbf{v}_0^k \int q(\textbf{v}_0^{S/k} | \textbf{v}_0^{k}, \textbf{v}_t^S) \ d\textbf{v}^{S/k}_0 \ d\textbf{v}^k_0 - \textbf{v}_t^k) $$
> $$ = \frac{\textbf{1}(k\in S)}{1-\bar\alpha_t} ( \sqrt{\bar \alpha_t} \int q(\textbf{v}_0^k | \textbf{v}_t^S) \textbf{v}_0^k \ d\textbf{v}^k_0 - \textbf{v}_t^k) $$
>
> We will add the intermediate steps in our revision to make it more accessible.
>
> ## Q3: Missing references of related works.
> Thanks for pointing it out. MAS and MVHuman are both interesting recent works that focus on 3D human/animal generation, which is a similar but different task from ours. We will add the references and discussion in our revision.

---

> > ### Comment · Reviewer_v3P6 · 2024-08-08
> > **Response to Rebuttal**
> >
> > I thank the authors' time and effort in addressing my concerns.
> >
> > I have carefully read all reviews and the rebuttal and am inclined to maintain my positive score for the submission.
> >
> > Please update the training and inference details in the final version as it seems most reviewers are confused.

---

### Official Review · Reviewer_EujL · 2024-07-09

**Soundness:** 3
**Presentation:** 3
**Contribution:** 3
**Rating:** 6
**Confidence:** 5

**Summary:**

This paper studies video generation with camera trajectories. The proposed method improves the consistency across multiple views via a cross-video synchronization module, which is equipped with the existing epipolar attention. Training of the proposed method consists of two phases: training to learn geometric consistency (3D) using ReadEstate10K and training with WebVid10M for learning motions. The authors utilized the augmentation methods such as video folding and homography augmentation for more effective learning addressing the scarcity of data. The proposed method achieves superior performance over baseline methods.

**Strengths:**

- **Strong performance**: The proposed pipeline achieves a significant performance gain compared to baselines: CameraCtrl and MotionCtrl. The camera control in video diffusion models is a relatively new topic and less explored in the literature. In both metrics, such as geometry and semantic consistencies, the proposed method outperforms baselines by a significant margin.
- **Comprehensive survey/related works**: This paper includes the most recent advances in video generation and discusses their strengths and weaknesses.
- **Analysis**: The appendix provides comprehensive analyses. First, the visualization of epipolar attention is included in the appendix. Also, qualitative results/generated videos (mp4) were helpful in understanding the quality of the generated videos.

**Weaknesses:**

- **Ablation study**: The data augmentations, video folding and homography augmentation, are applied. These augmentations are crucial to improve performance. The data augmentation strategies may be effective for the baselines CameraCtrl and MotionCtrl. However, due to the lack of ablation studies, it is hard to evaluate the impact of these components.
- **Lack of technical novelty**: Key value injection and epipolar attention have been proposed in previous works. Cross-view synchronization is merely a combination of existing techniques.
- **Qualitative analysis of trajectories**: Camera control is a challenging problem, and existing methods actually generate seemingly okay videos. However, the generated videos, especially when trajectories are unseen/new, do not precisely follow the input trajectories. Beyond AUC, camera pose comparisons will be helpful for readers to understand the quality of generated videos.

**Questions:**

- **Inference.** Multi-stage training has been proposed with/without additional parameters depending on datasets. It is not clear how to perform the inference. It needs a little bit more details.
- **Multiple views beyond two views and section 4.3.** How do you generate videos of multiple views beyond two views and Section 4.3 is not clear. What do you mean by M video features? Are they from different networks?
- **Editing.** Is it possible to apply the proposed pipeline to video editing? Given an input source video, is it possible to generate a new video with the same contents following two camera trajectories?

**Limitations:**

Limitations and Broader impacts are properly discussed in Section 6.1 and 6.2 The performance of the proposed method heavily relies on the performance of base video diffusion models. Also, because of the dependency, the proposed method cannot be applied to real-time applications, yet. Regarding negative societal impacts, the authors discussed deceptive content. The issue could be alleviated by better deepfake detectors.

---

> ### Author Rebuttal · Authors · 2024-08-07
>
> # Response of Reviewer EujL’s Review
>
> ## Q1: Lack of ablation study.
>
> We would like to emphasize that our cross-video augmentations are applied to the pairs of videos, which serve as the training input throughout our experiments; It is infeasible to apply such augmentations to monocular video generation methods such as MotionCtrl and CameraCtrl since they always only take one video as training input. To verify our design choices, including the effectiveness of our cross-video augmentations, we supply ablation study results on our method in Tab.3 in our supplementary, where our full model outperforms all ablative variants by a large margin, demonstrating the effectiveness of our design. For example, **our full model** achieves 25.2 scores at the Rot. AUC 5%, and scores of other variants are 16.8 for the **model without epipolar attention**; 17.9 for the **model without WebVid10M training**; 22.0 for the **model without homography transformation**;
>
> ## Q2: Qualitative Analysis of camera trajectories.
>
> In our paper, we show both quantitative and qualitative comparisons regarding camera trajectories. In Tab.1, the AUC scores evaluate the accuracy of camera poses optimized from our video predictions. In Fig.4, we show the results of our model and all baselines given the input camera trajectories (shown on the left). We think these experiments demonstrate the capability of our model to follow the input cameras.
>
> ## Q3: More details for inference (how to generate multiple views).
>
> Thanks for pointing this out. To generate more than two videos with our model, we designed a sampling algorithm as described in Section A.3.2. In general, the algorithm first initializes M noise maps from Gaussian (corresponding to the videos of M camera trajectories). In each denoising step, multiple pairs among the noise maps are selected, and the network is run on each selected pair for noise prediction. The predictions are then averaged over all pairs and applied to each noise map individually for denoising.
>
> Due to the page limit, we moved most of our inference details to our supplementary (Section A.3.2). If accepted, we will add more details to the main paper with the additional page quota.
>
> ## Q4: Is it possible to edit videos using our method (i.e., generate another video given a source video)?
>
> We think this is a very interesting and doable direction. While our model is trained to generate two videos simultaneously, it can also be adapted into a model conditioned on a source video. To do that, we can train it with the same training dataset but feed the model with one video as a reference and let it predict the other one.
>
> We also notice that a concurrent work from Hoorick et al. [1] also made a good attempt at this setting. We believe this demonstrates the great potential of multi-view video generation models to be applied in many applications in the future.
>
> ## Q5: Novelty of our paper.
>
> We believe our work has several significant distinctions from previous works. To our knowledge, we are the first work attempting the task of general multi-view video generation; We introduce the cross-view synchronization module (CVSM), which is inspired by previous works on epipolar attention, but has a very different approach aiming for the synchronization across videos in the task of multi-view video generation; And we propose a novel training-free algorithm to sample more consistent views from our model at inference.
>
> [1] Generative Camera Dolly: Extreme Monocular Dynamic Novel View Synthesis, Hoorick et al., 2024.

---

> ### Comment · Area_Chair_Yb4x · 2024-08-11
>
> Has the author's rebuttal addressed your concerns?
>
> Please provide your feedback on the rebuttal and make your final decision.
>
> AC

---

> > ### Comment · Reviewer_EujL · 2024-08-12
> >
> > I appreciate the authors' detailed responses. They have addresses all my minor concerns. Originally, I did not have any major concerns. I will keep my original rating.

---

### Official Review · Reviewer_MCcD · 2024-07-22

**Soundness:** 3
**Presentation:** 2
**Contribution:** 3
**Rating:** 5
**Confidence:** 5

**Summary:**

The paper introduces a framework to generate consistent multi-view videos of the same scene. Existing models lack precise camera control and consistency across views. The proposed CVD framework uses a cross-video synchronization module with an epipolar attention mechanism to maintain frame consistency from different camera trajectories. Trained with a combination of static multi-view and dynamic monocular data, CVD outperforms existing methods in generating consistent multi-view videos, as claimed by the authors.

**Strengths:**

- This work explores trajectory controllable multi-view video generation, while many of the existing works solely consider single trajectory generation. The research problem is a new setup.
- The authors propose a two-stage training strategy to learn multi-view camera motion in the real world. Though this may have some limitations, this is a new practical solution since there is a lack of large-scale multi-view video datasets with camera poses for dynamic scenes in the community.

**Weaknesses:**

- In Eq.(4), what does the x_i represent? Should it be x_1 here? I guess this is a typo. I also don’t find the notation illustration for x, which makes it hard to understand the purpose of using Eq.(4) to generate the attention mask. I suggest the authors to provide more details on this.
- For WebVid10M video dataset, the authors use homography warping for the training video generation. However, warping has many limitations. Technically, it only deals with planar points. Applying it to all video frames for “fake new view synthesis” will suffer from lost depth, perspective distortion, and incorrect parallax. Also, homography warping makes it hard to simulate new views from large rotations and translations. The visualized examples provided in the material also don’t show large camera motions. I notice that concurrent works usually construct a “real” multi-view dataset consisting of videos rendered from multi-view from 3D objects. Actually, the authors have a similar solution by re-organizing the videos from the multiview video dataset RealEstate10K and proposing the two-stage training, but the camera motion is still limited. Do the authors have some comments on this?
- The authors mentioned that “For the RealEstate10K dataset, we use CameraCtrl with LoRA fine-tuned on RealEstate10K as the backbone and applying the ground truth epipolar geometry in the cross-video module. For the WebVid10M dataset, we use AnimateDiff with LoRA fine-tuned on WebVid10M as the backbone, and apply the pseudo epipolar geometry (the same strategy used for the first frames in RealEstate10K dataset) in the cross-video module.” It seems that the authors are applying different model structures to train the model for the two datasets. In the inference, how do we combine these two separate models into one? Since the first one is supposed to learn static multiview geometry transform, the other one is supposed to learn more about video dynamics like effect or object motion. How do the learned priors combine together with consistency?

**Questions:**

See weakness above.

Justification for rating:

This paper proposes a new pipeline for a relatively new field, i.e., collaborative video diffusion for camera-controllable multi-view video generation. The results provided by the authors show some improvements compared to the baseline CameraCtrl. However, due to the questions and limitations I raised above, I give a slightly positive score here.

**Limitations:**

See above.

---

> ### Author Rebuttal · Authors · 2024-08-07
>
> # Response of Reviewer MCcD’s Review
>
> ## Q1: Clarification of Eq.(4) (how the attention masks are generated).
>
> Thanks for pointing this out. The attention masks are calculated as follows: For each pair of pixels coordinated at x_1 and x_2 in two frames, respectively, the attention mask from x_1 to x_2 is calculated by the epipolar distance between x_1 and x_2 (i.e. the shortest distance between x_1 and the epipolar line of x_2 in x_1’s frame). We will fix the typo and add the clarification in our revision.
>
> ## Q2: Limitations of homography warping and other possible dataset candidates.
>
> Thanks for raising this important question. Technically, the homography warping phase is introduced to enhance the model’s capability to produce synchronized motion across multi-view videos, which is an action taken out of necessity due to the lack of large-scale generic 4D datasets. Indeed, as the reviewer suggested, homography warping has many downsides, such as distortion of shape, incorrect perspective, and lack of view-dependent appearance and depth information. However, despite the limitations, homography warping serves as a pseudo-bridge attempting to close the gap between the training of static RealEstate10K videos and dynamic WebVid10M videos. Further, we argue that the perspective issue can be addressed by the pseudo epipolar lines: They are similarly distorted as the warped geometry, hence canceling out its effect on the model. In other words, it pushes the video model to strictly follow the given epipolar lines, making it eventually generate decent videos when the lines become correct. The combination of homography warping and epipolar-based inductive bias both effectively improves the transferring of learned correspondence information between our two training stages, as indicated in Tab.3 in our supplementary.
>
> In our work, we choose to follow CameraCtrl and MotionCtrl and use RealEstate10K, which has less diversity in camera trajectories since it only consists of monocular videos. But we believe our model can be further improved by integrating more datasets into the training, such as 3D/4D object datasets (Objaverse, OmniObject3D, MvImgNet), 3D landscape datasets (MegaScene, Acid), and small-scale 4D scenes (Multi-view video datasets). On the other hand, there exist many real-world dynamic motions that are hard to capture in multi-view, such as the motion of fireworks, waves, or wild animals. In these cases, monocular videos are still the only available large-scale resources. We believe there is great potential for exploring future works based on our method with additional datasets, and will add more discussions in the limitation section of our paper.
>
> ## Q3: How do you combine the models trained in different phases?
>
> As our model inherits the ability from AnimateDiff to generalize on different LoRAs, during inference we select the LoRA that best satisfies the needs in different settings. For the quantitative comparisons, we use LoRA trained on RealEstate10K (provided by CameraCtrl) for RealEstate10K scene generation and LoRA trained on WebVid10M (provided by AnimateDiff) for WebVid10M scene generation; For qualitative comparisons, without specification, we use the RealisticVision LoRA for its superior performance in generating high-quality images and videos. On the other hand, LoRA from CameraCtrl for temporal layers is applied in all experiments.

---

> > ### Comment · Reviewer_MCcD · 2024-08-08
> > **Reply to the rebuttal**
> >
> > Thanks to the authors for the clarification.
> >
> > - Thanks for the explanation on the warping. However, I still think that the current training curriculum and generated results lack enough ability to enable larger camera motion. The discussed solutions may work but they have been done and verified in this submitted version.
> > - I notice that the proposed method can't train a consistent and generalizable model to treat different scenarios. The proposed method needs to switch different LoRAs trained from datasets from different domains to generate different content for the best performance. This is a major flaw that seriously harms the generalization ability of the method.
> > - Also, I doubt whether the current model can handle diverse and distant camera trajectories for collaborative (or multi-view) video generation. Most of the given visualized examples show connected or spatially close camera trajectories.
> >
> > Considering the above concerns, I keep my final rating on the borderline.

---

> ### Author Response · Authors · 2024-08-09
> **Reply to Reviewer MCcD**
>
> We thank the reviewer for the reply and respectfully disagree with some of the claims.
>
> ## (Q1, Q3) Our model does not have enough ability to generate large-scale camera changes:
>
> **We first argue that our current model can handle large camera motions to some extent, as shown in our results**. For example, sample #3 in the first generation results in our demo video (0:01) are controlled by very different cameras; In the first demo of our 6-view video generation (1:35 in the demo video), the differences between the view on the left-top and the one on the bottom-right are also significant. These examples represent some of the most diverse and extreme camera trajectory variations possible that we can think of, within a 16-frame sequence, demonstrating that our method performs well even when the camera moves in opposite directions (e.g., flying in vs. flying out, rotating left vs. rotating right).
>
> Second, CVD is grounded in the currently limited infrastructures of open-source video generation and available datasets, functioning as a proof-of-concept that maximizes the potential of existing resources. This small-scale setup naturally affects the model’s capability to support large camera motions for two reasons:
> The capabilities of our method are inherently constrained by the performance of our base models, namely CameraCtrl. Consequently, the extent of motion and the range of camera paths we can manage are limited by the pretrained AnimateDiff and CameraCtrl models. Since the camera path conditioning is derived from RealEstate10K’s camera paths in CameraCtrl (and other models such as MotionCtrl, VD3D), our method cannot accommodate dramatic camera changes beyond what is covered in the RealEstate10K domain. As an extension of existing camera-conditioned video generation techniques, our approach necessarily inherits these limitations.
> At the time of our paper’s development, open-source video generation models were generally limited to producing only a few frames. For example, AnimateDiff, the base model used in our work, is capable of generating only 16-frame videos. Within this frame limit, accommodating large motions and significant camera movements is particularly challenging.
> Although CVD is relatively small in scale, we are inspired by our results and see significant potential in a scaled-up version incorporating systems like SORA or DreamMachine. We believe our work will inspire and motivate future research and development efforts in multi-view and multi-camera video generation.
>
>
>
>
> ## (Q2) Our model lacks generalization by using different LoRA for different experiments:
>
> We want to clarify that it is a common practice for diffusion models to apply different LoRA in different tasks. Both CameraCtrl and AnimateDiff are trained with specific LoRA to adapt the training data distribution, but are combined with unseen LoRAs in their qualitative results.
> While we apply different LoRAs for the spatial attention layers (which are pretrained models from previous works) for each task, our CVSM module remains the same across all experiments. We also want to emphasize that, although during the training our model **has never seen RealVision’s LoRA module**, it can naturally adapt this LoRA during the inference. These indicate that our CVSM can be generalized in various tasks.
>
> On the other hand, even though we have shown that our model generalizes well to unseen LoRAs, this does not mean our method does not work well without a LoRA. To support this, we also provide additional results without using any appearance LoRA (along with results in Table 1. in our paper):
>
> Model w/o LoRA:
>
> Rot. AUC: 21.9 / 34.6 / 49.4
>
> Trans. AUC: 2.7 / 7.4 / 16.3
>
> Prec.: 44.0
>
> M-S: 17.6
>
> Full Model:
>
> Rot. AUC: 25.2 / 40.7 / 57.5
>
> Trans. AUC: 3.7 / 9.6 / 19.9
>
> Prec.: 51.0
>
> M-S: 23.5
>
> MotionCtrl [56]+SVD:
>
> Rot. AUC: 12.2 / 28.2 / 48.0
>
> Trans. AUC: 1.2 / 4.9 / 13.5
>
> Prec.: 23.5
>
> M-S: 12.8
>
> Our model without any LoRA slightly underperforms our full model but still performs much better than the baselines.
>
> We also notice that merging different LoRAs into a single model is an active area of research (e.g. Work from Gu et al. [1] and Po et al. [2]). Therefore, while our setting matches standard practice in the field, we believe it would be interesting to explore CVD in combination with very recent advances, like MoS or orthogonal adaptation, which we will clarify in the discussion.
>
> [1] Mix-of-show: Decentralized low-rank adaptation for multi-concept customization of diffusion models, Gu et al. 2024
> [2] Orthogonal adaptation for modular customization of diffusion models, Po et al., 2024
>
> Thank you again for your time and consideration.

---

### Official Review · Reviewer_WFsL · 2024-07-22

**Soundness:** 2
**Presentation:** 3
**Contribution:** 2
**Rating:** 6
**Confidence:** 5

**Summary:**

This paper proposes a diffusion-based video generation method that generates multiple videos of the same scene simultaneously from camera trajectories and a text prompt. A cross-video synchronization module is proposed, where epipolar attention is introduced to improve the consistency across multiple videos. Experimental results show that the proposed method outperforms state-of-the-art approaches, especially in cross-video geometric consistency and semantic consistency.

**Strengths:**

1. The paper presents a new task that is to simultaneously generate videos of the same scene given multiple camera trajectories.

2. The paper is well-organized and easy to follow.

3.  A cross-video synchronization module is proposed to ensure cross-video geometric consistency.

**Weaknesses:**

1. When the proposed model is trained on different datasets (RE10K LoRA, CameraCtrl Pose LoRA, WV10M LoRA), different LoRA modules are used in the two-phase hybrid training. How are these different LoRA modules used in the final inference step? Are they used together, or is only one LoRA module chosen according to some settings?

2. Eq. 4 is confusing. The variables of Eq. 4 are x_i and x_2. However, x_i is not used on the right side of Eq. 4.

3. It's interesting to exploit the epipolar geometry to ensure cross-video geometric consistency. However, some technical details are unclear in Sec. 4.1. It is not very clear how the epipolar geometry is used to generate the attention mask M in Eq. 4.  It would be better if the authors provided more details and more explanations. Otherwise, it is difficult to reproduce the proposed method.

4. The attention mask is additionally introduced in the cross-view synchronization module. Yet, it is confusing why this mask is being introduced. Does the attention mask indicate correspondences between frames  of different videos?  It would better to provide some statistical results (average percentage ) for the attention mask.

5. Given a video diffusion model perfectly ensures temporal coherence and has excellent camera control performance， is it still necessary to generate multiple videos simultaneously? Can we use the video diffusion model to generate one video at a time instead?

**Questions:**

Please refer to my comments above

**Limitations:**

The authors have addressed the limitations and potential negative societal impact of their work.

---

> ### Author Rebuttal · Authors · 2024-08-07
>
> # Response of Reviewer WFsL’s Review
>
> ## Q1: How are the LoRA modules used in the inference step?
>
> As our model inherits the ability from AnimateDiff to generalize on different LoRAs, during inference we select the LoRA that best satisfies the needs in different settings. For the quantitative comparisons, we use LoRA trained on RealEstate10K (provided by CameraCtrl) for RealEstate10K scene generation and LoRA trained on WebVid10M (provided by AnimateDiff) for WebVid10M scene generation; For qualitative comparisons, without specification, we use the RealisticVision LoRA for its superior performance in generating high-quality images and videos. LoRA from CameraCtrl for temporal layers is applied in all experiments.
>
> ## Q2: Wrong notation in Eq. 4. (x_i should be x_1).
>
> Thanks for pointing this out. We will fix this typo in our revision.
>
> ## Q3: Clarification on how the attention masks are generated and used.
>
> Our attention masks are added as an inductive bias to the self-attention modules, encouraging them to extract information from the epipolar geometry. For this reason, our attention masks function as a 0/1 masking on the full self-attention score, where only the pixels on the corresponding epipolar lines have attention weights, as shown in Fig.6 of our supplementary. This process does not directly indicate correspondences (since we never possess ground truth pixel correspondences during training and inference) but rather introduces the epipolar line inductive bias about where to find the correspondences. As shown in our ablation study in Tab.3, having this inductive bias is very helpful in terms of aligning the content across the two frames. Empirically, we find the hit rate of a pixel’s corresponding pixels being picked up by the attention module (at some diffusion step) close to 100%.
>
> ## Q4: Comparison between our model and running monocular video diffusion model multiple times.
>
> Indeed, running a powerful monocular VDM multiple times with the same condition (ideally the same keyframes) is a rather reasonable method for multi-view video generation. Yet, this approach faces a major issue: It will suffer from generating inconsistent motion and different content that is unseen in the keyframes. The reason is that monocular VDMs are not trained to introduce consistency across views. For example, if we let a monocular VDM generate multiple firework videos given the same condition (text prompt or starting frame), it will very likely generate different firework patterns in each pass. Our model instead jointly predicts all videos together with their information being shared in the cross-view modules, thus it can produce more view-consistent results. In our experiments (Tab.2), we compare our model with the baseline “CameraCtrl+SparseCtrl”, which can be considered as the monocular VDM approach. Our model outperforms the latter by a large margin.

---

> > ### Comment · Reviewer_WFsL · 2024-08-11
> >
> > Thank the authors for answering my questions.  I still have questions about whether the proposed method can ensure the Synchronization of generated multi-videos (also the Cross-View Synchronization Module).  The concern is from the training data.  As stated in the paper, the paper constructs a pair of training data by first sampling 2N − 1 frames from a video in the dataset and divides them into two clips (e.g., N,-1 ) from the middle, where the first part is reversed.  There is no guarantee that the content of the i-th frames in both video clips is synchronized or consistent. For example, a chair may appear in the first part and disappear in the second part.

---

> > > ### Author Response · Authors · 2024-08-12
> > >
> > > Dear Reviewer,
> > >
> > > Thank you for your insightful feedback and for recognizing the value of our proposed method. We appreciate the opportunity to clarify the concerns regarding the synchronization of generated multi-videos and the training of our Cross-View Synchronization Module (CVSM).
> > >
> > > We train our CVSM module using two data sources:
> > > - RealEstate10K videos, which are static scene captures.
> > > - WebVid10M videos, which are dynamic in-the-wild videos.
> > >
> > > As outlined in our Sec. 4.2, **the middle-cut and reverse scheme is only applied to the static RealEstate10K videos** — the static nature of these videos ensures that epipolar geometry always holds, therefore the above-mentioned scenario, where a chair moves (e.g. appears in the first part but disappear in the corresponding frame in the second part) will never happen, as there is no dynamic movement in these training videos. The only possibility that such a scenario may arise is if the camera itself moves beyond the visibility of the chair, which is commonly presented in our training data.  In this scenario, the content shared across two views would be synchronized by our epipolar attention in CVSM. For the non-overlapping regions, consistency becomes trivial and it typically relies more on the model’s generation capability to construct new content. Our model does not alter the parameters of our base video generative models and is trained with both RealEstate10K and WebVid10M to handle generation effectively, with generation performances on par with our baselines as indicated in Tab. 2. **For the dynamic WebVid10M videos, we intentionally avoid applying the middle-cut and reverse scheme** to prevent the exact issues you highlighted. Instead, we create paired data by duplicating a clip and applying homography augmentation. In this case, the paired videos are perfectly synchronized, as they are essentially copies of each other with perfectly consistent timestamps.
> > >
> > > Therefore, our CVSM module, derived from epipolar geometry, is always trained on paired videos that hold perfect epipolar geometry and are therefore always perfectly synchronized. This ensures that as a generative model, the output of our model is also synchronized.

---

> > > > ### Comment · Reviewer_WFsL · 2024-08-13
> > > >
> > > > Thank the authors for answering my questions.
> > > > - RealEstate10K videos.  Some videos in this dataset have larger camera motions. Could it be that the i-th frames of the two videos have only a few corresponding regions?
> > > > - WebVid10M videos. The paper uses random homography transformations to create video pairs.  Does this restrict the difference in camera trajectories between the two videos from being too significant
> > > >
> > > > In addition,  the fireworks are not so synchronized, as shown in the last two rows of Fig. 13 and the demo video

---

> > > > > ### Author Response · Authors · 2024-08-13
> > > > >
> > > > > We thank the reviewer for the reply and response the new issues below.
> > > > >
> > > > > ## Q1: Non-overlapping regions due to large camera changes.
> > > > >
> > > > > Yes, this is indeed possible and composes only a small portion of our training data. In this case, consistency becomes trivial as there is limited overlapping content, but our model is still fully compatible since it can simply generate high-quality new content for the non-overlapping regions too. This is because our model does not alter the parameters of our base video generative models and is trained with both RealEstate10K and WebVid10M to handle generation effectively, with generation performances on par with our baselines as indicated in Tab. 2. To summarize, our model can handle large camera motion and non-overlapping scenarios with no problem, so the non-overlapping large camera movement videos in our training set do not affect performances of our model.
> > > > >
> > > > > ## Q2: Camera Differences in Homography Transformation.
> > > > >
> > > > > We, in fact, adjust the homography transformation parameter according to RealEstate10K's camera movements so that it's a relatively close simulation. Therefore, whatever we have for the homography augmentation should be similar to the camera trajectories in RealEstate10K and will not restrict the difference in camera trajectories.
> > > > >
> > > > > ## Q3: Fireworks are not so Synchronized.
> > > > >
> > > > > As we mentioned in our limitation section, our method is heavily dependent on the performances of our base models. Since we built CVD on AnimateDiff, which has quite a lot of flicking and content shifting itself, our results are potentially influenced too and are certainly not perfect --- we may suffer from the same type of issues that cause some small inconsistency and non-synchronized parts. We believe with newer, more consistent video generation models with fewer flickings such as OpenSORA/CogVideoX, our method should transfer, and potentially, the small issues will address themselves.

---

> > > > > > ### Comment · Reviewer_WFsL · 2024-08-14
> > > > > >
> > > > > > Thank the authors for the discussion and for addressing most of my concerns. It may be better to provide more details of the presented datasets.   I will keep my positive score.

---

### Official Review · Reviewer_3sRf · 2024-07-25

**Soundness:** 2
**Presentation:** 2
**Contribution:** 2
**Rating:** 5
**Confidence:** 4

**Summary:**

The paper proposes a novel framework that generates multi-view videos from text input. The model builds upon the CameraCtrl pipeline and includes a Cross-View Synchronization Module to enforce consistency guided by a fundamental matrix.

**Strengths:**

- The performance of the proposed method is impressive, generating multi-view videos from text input.
- The framework leverages both static videos and dynamic videos for joint training.

**Weaknesses:**

- Random homography transformations are applied to the clones for WebVid videos. It would be great if examples of the augmented clones were visualized. I'm mainly worried that the augmented clones seem to introduce a lot of black (unknown) regions and how much it affects the performance of video generation.
- Table 2 should include video-based metrics, such as FVD.

**Questions:**

- How is the 3D reconstruction (shown in supp_video.mp4) obtained using the generated multi-view videos?
- How is Table 3 obtained? Why does training on WebVid10M (no camera pose) improve Rot. AUC Trans. AUC (camera accuracy) as compared to training on RE10K only? I'm confused by the results in Table 3.

**Limitations:**

Please refer to the weakness.

---

> ### Author Rebuttal · Authors · 2024-08-07
>
> # Response of Reviewer 3sRF’s Review
>
> ## Q1: The black regions might affect the performance of video generation.
>
> During our training, we removed the L2 loss in the unseen pixels (black regions) of the cloned video for data integrity. We show examples of homography warped videos in our attached PDF (Fig. A2) and will clarify the loss in black regions in our revision.
>
> ## Q2: Add FVD metric evaluation.
>
> Thanks for pointing this out. We evaluate our method and all baselines using the FVD metric, and here are the results:
> - CameraCtrl: 277
> - AnimateDiff+SparseCtrl: 327
> - CameraCtrl+SparseCtrl: 430
> - Ours: 285
>
> Similar to our experiments on the FID metric, our method is on par with CameraCtrl and performs better than other baselines.
>
> ## Q3: How is the 3D reconstruction obtained using the generated multi-view videos?
>
> We use NeRFactor [Tancik et al. 2023] to reconstruct 3D from our multi-view videos. We will clarify more details in the revised version.
>
> ## Q4: How is Tab.3 obtained? Why does training on WebVid10M improve Rot. AUC Trans. AUC as compared to training on RealEstate10K only?
>
> This is an insightful question! For the evaluation results in Tab.3, each model generates video pairs given the same text prompt and camera trajectories. For each video pair, we calculate the SuplerGlue [Sarlin et al. 2020] matches between each frame pair (i.e. two frames captured at the same time from the video pair). Then, we run RANSAC on these matches to calculate the relative camera poses between the two frames in each frame pair and compare the camera poses with the ground truth to get the AUC score.
>
> We believe there are two reasons why our full model outperforms the model trained on RealEstate10K. Firstly, we think the credit goes to our epipolar attention. In the WebVid10M training stage, while there are no camera poses available, we managed to use pseudo-gt epipolar lines (i.e. lines calculated from homography matrix H. The line of pixel x in the warped frame goes through the pixel Hx) to describe the spatial relationship between video frames. This enhances the model’s ability to generate videos that satisfy the given line conditions. Hence, in a camera-control setting, the full model is more constrained to the epipolar lines and generates videos that align better with the camera poses. Secondly, since RealEstate10K mostly consists of static indoor scenes, models trained on RealEstate10K may suffer from data bias and may not perform well on general scenes,  thus resulting in poor evaluation performance in this experiment. We will clarify this in our revision.

---

> ### Comment · Area_Chair_Yb4x · 2024-08-11
>
> Has the author's rebuttal addressed your concerns?
>
> Please provide your response to the rebuttal and make your final decision.
>
> AC

---

### Author Rebuttal · Authors · 2024-08-07

We appreciate the thorough review and constructive feedback provided on our work. We are happy to see that the reviewers recognize our work as a novel attempt at a new and interesting task (Reviewer **WFsL, MCcD, v3P6**), our model is interesting (Reviewer **WFsL**) and achieves strong performance (Reviewer **3sRf, EujL, v3P6**), and our paper is well-written (Reviewer **WFsL, v3P6, EujL**). We will fix all typos and missing references in our revision.

We want to emphasize that our work aims to solve the very new task of multi-view video generation. Compared to other generative models for images, videos, and 3D objects, this task is much more challenging due to the lack of large-scale multi-view video datasets.
By introducing this new problem setting, we aim to encourage further exploration of how video generation models can be utilized for future scene-level dynamic 3D and 4D generation tasks.
With the rapid progress of large-scale video diffusion models, we strongly believe the multi-video generative model has great potential to evolve and benefit many downstream applications, such as immersive content creation, video editing, and communications.

We incorporate some fruitful suggestions from the reviewers and provide additional results in this rebuttal:
- Comparison with baselines on the FVD metric: Our model is on par with CameraCtrl and outperforms other baselines. This matches the results on FID and KID in our paper as well.
  - CameraCtrl: 277
  - AnimateDiff+SparseCtrl: 327
  - CameraCtrl+SparseCtrl: 430
  - Ours: 285
- Additional qualitative comparisons on new camera trajectories (Fig.A1 in PDF)
- Visualization of the homography warping applied in our WebVid10M phase. (Fig.A2 in PDF).

Here we address some of the most common issues.

## Q1: Clarification for the epipolar attention mechanism (Eq. 4).

The epipolar attention mechanism is added to provide inductive bias from input camera poses to the self-attention modules, encouraging them to extract information from the epipolar geometry. Specifically, it incorporates attention masks that function as a 0/1 masking on the full self-attention score, where only the pixels on the corresponding epipolar lines have attention weights, as shown in Fig.6 of our supplementary. Eq. 4 shows how the attention mask is calculated: For each pair of pixels located at x_1 and x_2 in the two input frames, respectively, the attention mask from x_1 to x_2 is determined by their epipolar distance (i.e. how far is it from x_1 to the epipolar line of x_2 in x_1’s frame). We will fix the typo (x_i -> x_1) in Eq. (4) and add more explanation in our revision.

## Q2: More details of our training phases.

The two training phases are applied to the same instance of CVSM in a single training process. Specifically, we load the LoRAs in both phases to the pipeline, and then, for each training step, a data batch from the joint of Webvid10M (MV10M) and RealEstate10K is sampled. The corresponding LoRAs are activated for training based on which dataset the data comes from.

## Q3: Clarification of the LoRA selection during inference.

Different LoRA selecting strategies are applied to the Stable Diffusion layers (i.e. Spatial attention layers) and the AnimateDiff layers (i.e. Temporal attention layers). For Stable Diffusion layers, as our model inherits the capability of AnimateDiff to adapt various LoRAs, we use different LoRAs that best fit the task in different experiments. In the quantitative experiments, RealEstate10K’s LoRA is used for RealEstate10K’s test case, and WebVid10M’s LoRA is used for WebVid10M’s test case; In the qualitative experiments, without specification, RealisticVision’s LoRA is applied for its great performance to generate high-quality images/videos. For temporal attention layers, the LoRA from CameraCtrl is used in all experiments.

---

### Decision · Program_Chairs · 2024-09-25

**Decision:**

Accept (poster)

**Comment:**

The reviewers raised concerns mainly about the training strategy's clarity, the mathematical derivations' correctness, the integration and generalization of LoRA modules during inference, and the lack of ablation studies to evaluate the impact of key components. They also suggested including recent related works on consistent multi-view content generation.

The authors responded comprehensively by clarifying the training phases, explaining the epipolar attention mechanism and LoRA selection process, addressing the derivation issues, and providing additional comparisons and qualitative results. They also acknowledged the need for more discussions on related works and ablation studies, which they committed to addressing in their revision.

The ACs agree that the authors have adequately addressed the concerns, and the paper should be accepted.